# CARSO: BLENDING ADVERSARIAL TRAINING AND PURIFICATION IMPROVES ADVERSARIAL ROBUSTNESS

## ABSTRACT

In this work, we propose a novel adversarial defence mechanism for image classification – CARSO – blending the paradigms of *adversarial training* and *adversarial purification* in a mutually-beneficial, robustness-enhancing way. The method builds upon an adversarially-trained classifier, and learns to map its *internal representation* associated with a potentially perturbed input onto a distribution of tentative clean reconstructions. Multiple samples from such distribution are classified by the adversarially-trained model itself, and an aggregation of its outputs finally constitutes the *robust prediction* of interest. Experimental evaluation by a well-established benchmark of varied, strong adaptive attacks, across different image datasets and classifier architectures, shows that CARSO is able to defend itself against foreseen and unforeseen threats, including adaptive *end-to-end* attacks devised for stochastic defences. Paying a tolerable *clean* accuracy toll, our method improves by a significant margin the *state of the art* for CIFAR-10 and CIFAR-100 $\ell_\infty$ robust classification accuracy against AUTOATTACK.

## 1 INTRODUCTION

Vulnerability to adversarial attacks (Biggio et al., 2013; Szegedy et al., 2014) – *i.e.* the presence of specific inputs, usually crafted on purpose, able to catastrophically alter the behaviour of high-dimensional models (Bortolussi & Sanguinetti, 2018) – constitutes a major hurdle towards ensuring compliance of deep learning systems with the behaviour expected by modellers and users, and their adoption in safety-critical scenarios or tightly-regulated environments. This is particularly true for adversarially-*perturbed* inputs, where a norm-constrained perturbation – often hardly detectable by human inspection (Qin et al., 2019; Ballet et al., 2019) – is added to an otherwise legitimate input, with the intention of eliciting a potentially malicious anomalous response (Kurakin et al., 2018).

Given the pervasiveness of the issue (Ilyas et al., 2019), and the serious concerns raised about safety and reliability of data-learned models in the lack of appropriate mitigation (Biggio & Roli, 2018), adversarial attacks have been extensively studied. Yet, that of obtaining generally-robust machine learning (*ML*) systems remains a longstanding issue, and a major open challenge.

Research in the field has been animated by two opposing (yet complementary) efforts. On the one hand, the study of *failure modes* in existing models and defences, with the goal of understanding their origin and developing stronger attacks with varying degrees of knowledge and control over the target system (Szegedy et al., 2014; Goodfellow et al., 2015; Moosavi-Dezfooli et al., 2016; Tramèr et al., 2020). On the other hand, the construction of increasingly capable defence mechanisms. Though alternatives have been explored (Cisse et al., 2017; Tramèr et al., 2018; Carbone et al., 2020; Zhang et al., 2022), most part of the latter is based on adequately leveraging *adversarial training* (Goodfellow et al., 2015; Madry et al., 2018; Tramèr & Boneh, 2019; Rebuffi et al., 2021; Gowal et al., 2021; Jia et al., 2022; Singh et al., 2023; Wang et al., 2023; Cui et al., 2023; Peng et al., 2023), *i.e.* training a *ML* model on a dataset composed of (or enriched with) adversarially-perturbed inputs associated with their correct, *pre-perturbation* labels. And indeed, adversarial training has been the only technique so far able to consistently provide an acceptable level of defence (Gowal et al., 2020), while still incrementally improving up to current *state of the art* (Cui et al., 2023; Peng et al., 2023).

Another defensive approach is that of *adversarial purification* (Shi et al., 2021; Yoon et al., 2021), where a generative model is used – akin to denoising – to recover a perturbation-free version of the input before classification occurs. Nonetheless, such attempts have generally fallen short of

expectations due to inherent limitations of the generative models used, or due to decreases in robust accuracy[1] when attacked *end-to-end* (Gu & Rigazio, 2015) – resulting in subpar robustness if the defensive structure is known to the adversary (Tramèr et al., 2020). More recently, the rise of diffusion-based generative models (Huang et al., 2021) and their use for purification enabled more successful results (Nie et al., 2022; Chen et al., 2023) – though at the cost of lengthy inference and training times.

In this work, we draw inspiration from neurocognitive processes underlying *cued recall* and *recognition* (Tulving & Thomson, 1973; Eich, 1980; Medina, 2008) to devise a novel adversarial defence for supervised image classification, dubbed CARSO (Counter-Adversarial Recall of Synthetic Observations). The approach relies on an adversarially-trained classifier (called hereinafter simply *the classifier*), endowed with a generative stochastic model (called hereinafter *the purifier*). The latter learns to generate – from the ordered tensor[2] of activations registered at neuron level in the former, upon classification of a potentially-perturbed input – samples from a distribution of plausible, perturbation-free reconstructions. At inference, a (numerous) sample of such reconstructions is classified by the very same *classifier*, and the original input robustly labelled by aggregating its outputs. Such method – to the best of our knowledge the first attempt to organically merge the *adversarial training* and *purification* paradigms – avoids the vulnerability pitfalls typical of the mere stacking of a purifier and a classifier (Gu & Rigazio, 2015), while still being able to take advantage of individual improvements to its parts (*i.e.* in adversarial training or generative modelling, independently).

Empirical assessment[3] of the defence in the $\ell_\infty$ *white-box* setting is also provided – using a *conditional* (Sohn et al., 2015; Yan et al., 2016) *variational autoencoder* (Kingma & Welling, 2014; Rezende et al., 2014) as purifier, and existing *state of the art* (or otherwise notable) pre-trained models as classifiers. Such choices are meant to give to existing approaches – and to the *adversary* attacking our architecture *end-to-end* as part of evaluation – the strongest advantage possible. Yet, in all scenarios considered, CARSO improved significantly upon the robustness of the adversarially pre-trained classifier – including in case of attacks specifically devised to fool stochastic defences. Remarkably, with tolerable *clean* accuracy toll, our method also improves by a significant margin the current *state of the art* for CIFAR-10 (Krizhevsky, 2009) and CIFAR-100 (Krizhevsky, 2009) robust classification accuracy against AUTOATTACK (Croce & Hein, 2020a).

In summary, the paper makes the following contributions:

- CARSO, a novel adversarial defence method synergistically blending *adversarial training* and *purification*;

- A *bag of tricks* to make possible and ease the training of such architecture, when the *purifier* is a *conditional variational autoencoder* – but potentially applicable to other scenarios as well;

- Experimental assessment of the method proposed, against standardised benchmark adversarial attacks – showing higher robust accuracy *w.r.t.* to existing *state of the art* adversarial training or purification approaches, and defying unforeseen *end-to-end* attacks.

The rest of the manuscript is structured as follows. In section 2 we provide an overview of specific contributions in the fields of *adversarial training* and *purification-based* defences – with focus on image classification. In section 3, a deeper analysis is given of two integral parts to our experimental assessment: PGD adversarial training, and (conditional) variational autoencoders. Section 4 is dedicated to the intuition behind CARSO, its architectural description, and the *tricks* used during its training. Section 5 contains details about the experimental setup, results, and comments. Section 6 concludes the paper and outlines directions of future development.

---

[1] The *test set accuracy* of the frozen-weights trained classifier – computed on a dataset entirely composed of adversarially-perturbed examples, generated against the model.

[2] Which we call *internal representation*.

[3] Implementation of the method and code for the experiments (based on *PyTorch* (Paszke et al., 2019), `AdverTorch` (Ding et al., 2019), `TorchAttacks` (Kim, 2020), and `ebtorch` (Ballarin, 2023)) can be found in Supplementary Materials.

## 2 RELATED WORK

***Adversarial training* as a defence**    The idea of training a model on adversarially-generated examples as a way to make it more robust can be traced back to the very beginning of research in the area. The seminal work by Szegedy et al. (2014) proposes to perform training on a mixture of *clean* and adversarial data, generated beforehand.

The introduction of FGSM (Goodfellow et al., 2015) enables the efficient generation of adversarial examples during training – with a single normalised gradient step. Its iterative evolution PGD (Madry et al., 2018) – discussed in section 3 – improves significantly the effectiveness of adversarial examples produced, making it still the *de facto* standard for the synthesis of adversarial training inputs (Gowal et al., 2020). Further incremental improvements have also been developed, some focused specifically on robustness assessment (*e.g.* adaptive-stepsize variants, as in Croce & Hein (2020a)).

Most recent adversarial training protocols further rely on synthetic data to increase the numerosity of training datapoints (Gowal et al., 2021; Rebuffi et al., 2021; Wang et al., 2023; Cui et al., 2023; Peng et al., 2023), and adopt *tweaked* loss functions to balance robustness and accuracy (Zhang et al., 2019a) or generally foster the learning process (Cui et al., 2023). The entire model architecture may also be tuned specifically for the sake of robustness enhancement (Peng et al., 2023). Such ingredients are often required to reach the current *state of the art* in robust accuracy.

***Purification* as a defence**    Among the first attempts of *purification-based* adversarial defence, Gu & Rigazio (2015) investigate the use of denoising autoencoders (Vincent et al., 2008) to recover examples free from adversarial perturbations. Despite its effectiveness in the denoising task, the method may indeed *increase* the vulnerability of the model when attacks are generated against it *end-to-end*. The improvement proposed in such latter work (Gu & Rigazio (2015)) adds a smoothness penalty to the reconstruction loss, mitigating such downside. Similar in spirit, Liao et al. (2018) tackle the issue by computing reconstruction loss between the last-layers representations of the (frozen-weights) attacked classifier, receiving respectively as input the *clean* and the tentatively *denoised* example.

In Samangouei et al. (2018), *Generative Adversarial Networks* (GANs) Goodfellow et al. (2014) learned on *clean* data are used at inference time to find a plausible synthetic example – close to perturbed input – belonging to the unperturbed data manifold. Despite positive results, the delicate training process of GANs and the existence of known failure modes (Zhang et al., 2018) affect the method. More recently, a similar approach (Hill et al., 2021) employing *energy-based models* (LeCun et al., 2006) suffered from poor sample quality.

Purification approaches based on (conditional) variational autoencoders include Hwang et al. (2019) and Shi et al. (2021).

Finally, already-mentioned techniques relying on *score-* (Yoon et al., 2021) and *diffusion-* based (Nie et al., 2022; Chen et al., 2023) models have also been proposed, with surprisingly favourable results – often balanced in practice by longer inference and training times.

## 3 PRELIMINARIES

**PGD adversarial training**    The task of finding model parameters robust to adversarial perturbations is framed by Madry et al. (2018) as a *min-max* optimisation problem seeking to minimise *adversarial risk*. The inner optimisation (*i.e.* the generation of worst-case adversarial examples) is solved by an iterative algorithm – *Projected Gradient Descent* – interleaving gradient ascent steps in input space with the eventual projection on the border of an $\epsilon$-ball centred around an input datapoint.

In this manuscript, we will use the shorthand notation $\epsilon_p$ to denote $\ell_p$ norm-bound perturbations of maximum magnitude $\epsilon$.

Formal details of the method are provided in Appendix A.

**(Conditional) Variational Autoencoders**    Variational autoencoders (*VAEs*) (Kingma & Welling, 2014; Rezende et al., 2014) allow the learning, from data, of approximate generative latent-variable models of the form $p(\boldsymbol{x}, \boldsymbol{z}) = p(\boldsymbol{x} \mid \boldsymbol{z})p(\boldsymbol{z})$, whose likelihood and approximate posterior are para-

meterised by deep artificial neural networks (*ANN*s). The problem is cast as the maximisation of a variational lower bound.

In practice, optimisation is performed iteratively – on a loss determined by the mixture of reconstruction loss and empirical *KL* divergence *w.r.t.* the prior, computed on minibatches of data.

*Conditional* Variational Autoencoders (Sohn et al., 2015; Yan et al., 2016) extend *VAE*s by concatenating a *conditioning vector* $c$ – expressing specific characteristics of each example – to $z$ during training. This allows the learning of a decoder model capable of conditional data generation.

Further details on the functioning of such models are given in Appendix B.

## 4 DEVELOPMENT AND STRUCTURE OF CARSO

The core ideas informing the design of our method are driven more by *first-principles* and *analogical* reasoning rather than arising from specific formal requirements. This section is dedicated to the discussion of such ideas, the specification of the architectural details of CARSO, and to practical enhancements to its training process.

### 4.1 ARCHITECTURAL OVERVIEW AND PRINCIPLE OF OPERATION

From a purely architectural viewpoint, CARSO is composed of two *ANN* models – the already mentioned *classifier* and *purifier* – operating in close synergy. The only requirement of the former is that of having been *adversarially* trained to solve the classification task of interest – thus allowing free reuse of pre-trained models, and retaining contributions to overall robustness from established adversarial training techniques.

The key element of novelty lies in the intertwined operation of *classifier* and *purifier* – aimed at learning a distribution of purified examples (to be generated from each potentially-perturbed input) from the internal representation of the *classifier*, and to classify some samples harvested from it – without increasing (and indeed much reducing) the overall adversarial vulnerability in the process.

The *purifier* is also independent from specific architectural choices, provided it can sample, during inference, multiple (different) tentative reconstructions of the input conditionally on the internal representation of the classifier *alone* – *i.e.* without requiring further datapoint-dependent information. In the rest of the paper, we adopt a *conditional variational autoencoder* as the purifier of choice, receiving the internal representation of the *classifier* as *conditioning set* and operating decoder-only during inference. Such choice was due to its light computational training requirements and exact algorithmic differentiability (Baydin et al., 2018). The latter condition ensures that the *purifier* does not contribute to gradient obfuscation (Athalye et al., 2018a) when *end-to-end* adversarial attacks are produced against the whole architecture.

A diagram of the overall architecture is shown in Figure 1, and its detailed principles of operation described below.

**Training** During training, adversarially-perturbed examples are generated against, and fed to, the *classifier*. The tensor of *classifier* activations across all layers (in arbitrary but fixed order) is then extracted. At this point, the conditional *VAE* is trained on denoised input reconstruction as customary, conditioned on their corresponding previously extracted *internal representations*.

Upon completion of the training process, the encoder network may be discarded – as it will not be used for inference.

**Inference** The example requiring classification is fed to the pre-trained *classifier*. Its corresponding internal representation is extracted and used to condition the generative process described by the decoder of the *VAE* – whose stochastic latent variables are sampled from the original priors. Each element in the resulting set of denoised examples is classified by the same pre-trained *classifier*, and individually predicted classes are aggregated. The result of the aggregation is the robust prediction of the input class.

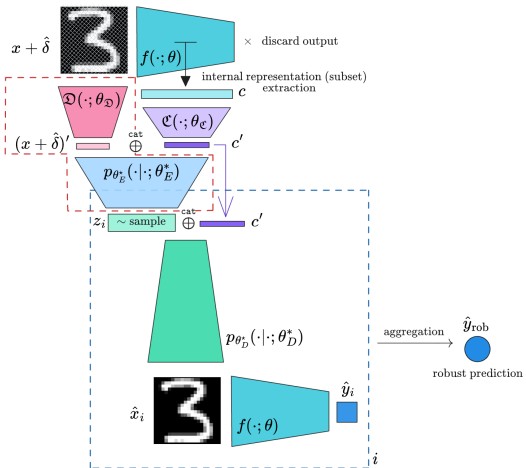

Figure 1: Schematic representation of the CARSO architecture used in the experimental phase of this work. The subnetwork bordered by the red dashed line is used only during training of the *purifier*. The subnetwork bordered by the blue dashed line is re-evaluated on different samples of $z$ and the resulting individual $\hat{y}_i$ aggregated into $\hat{y}_{\text{rob}}$. The *classifier* $f(\cdot; \theta)$ is always kept frozen; the remaining network is trained on $\mathcal{L}_{\text{VAE}}(x, \hat{x})$. Precise details on the functioning of the networks are provided in subsection 4.1.

Remarkably, the only link between the initial potentially-perturbed input and the resulting purified reconstructions (and thus, predicted class) is through the *internal representation* of the classifier – that serves as a *featurisation* of the original input. The whole process is also exactly differentiable, *end-to-end*.

## 4.2 A FIRST-PRINCIPLES JUSTIFICATION

If we consider a trained *ANN* classifier, subject to a successful adversarial attack by means of a slightly perturbed example, we observe that – both in terms of $\ell_p$ magnitude and human perception – a small variation on the input side of the network is amplified to a significant amount on the output side, thanks to the layerwise processing by the model. Given the deterministic nature of such processing at inference time, we conclude that the *trace* obtained by sequentially collecting activation values within the network, along the forward pass, constitutes a richer characterisation of such amplification process compared to input alone. And indeed, it is possible to learn a direct mapping from such featurisation of the input, to a distribution of possible perturbation-free input reconstructions – as explained earlier – that takes advantage of such additional knowledge.

Furthermore, we claim that using the same *classifier* for both *internal representation* extraction and classification of reconstructed inputs has a positive effect in shielding the entire architecture from gradient-based *end-to-end* attacks. Indeed, if one considers the total gradient arising at the level of individual neurons in the *classifier* – when an attack is performed against the whole architecture – it receives contributions from the two roles played at once by those neurons: as conditioning set for the *VAE*, and as part of the *classifier* architecture. Since the individual direction of those contributions is *a priori* not aligned, it is likely that – in expectation – the resulting gradient direction is not aligned with that of the gradient of the unshielded classifier, and its magnitude in such direction decreased.

## 4.3 EXAMPLE AND INTERNAL REPRESENTATION PRE-ENCODING

Given the high variability of datasets and classifier architectures across instances of robust classification – one potential limitation of the approach described so far comes from the heterogeneity in relative size and structure between data $x$ to be modelled and their corresponding conditioning set $c$. Indeed, the training of a conditional *VAE* requires (Sohn et al., 2015) concatenation of the two, and further concatenation of the latter with a sample of stochastic latent variables $z \sim q_{\theta_{\text{E}}}(z \mid x; \theta_{\text{E}})$.

Additionally, the *a priori* lack of spatial structure in $c$ strongly limits the use of *convolutional neural networks* (*CNN*s) (LeCun et al., 1998) within the encoder, whereas the use of *fully-connected networks* (*FCN*s) on images is deemed unsatisfactory – given the hardness of learning locally-convolutional structures from data (Ingrosso & Goldt, 2022).

Mitigating both issues, we propose to insert in the architecture two additional *auxiliary* encoders – again parameterised by *ANN*s – to be learned jointly with the *VAE* during training on the purification task: one relative to data $\mathfrak{D}(\cdot; \boldsymbol{\theta}_{\mathfrak{D}})$ and one to the *context* $\mathfrak{C}(\cdot; \boldsymbol{\theta}_{\mathfrak{C}})$.

The resulting *encoded* examples $\boldsymbol{x}' := \mathfrak{D}(\boldsymbol{x}; \boldsymbol{\theta}_{\mathfrak{D}})$ and context $\boldsymbol{c}' := \mathfrak{C}(\boldsymbol{c}; \boldsymbol{\theta}_{\mathfrak{C}})$ may be used in place of $\boldsymbol{x}$ and $\boldsymbol{c}$ respectively, within the *VAE*. This allows to effectively decouple data- and context-specific processing before *autoencoding* occurs: in the design of $\mathfrak{D}$ and $\mathfrak{C}$, the $|c|/|x|$ balance is ensured by tuning output sizes of the auxiliary encoders; structural heterogeneity is addressed by choosing the inductive biases (Mitchell, 1980) deemed the most suitable to the respective inputs.

In order to sample from the generative model at inference time, $\mathfrak{C}$ must be preserved after training, and used to encode the *internal representations* extracted. $\mathfrak{D}$ may instead be discarded.

## 4.4 ADVERSARIALLY-BALANCED BATCHES

Calling $\epsilon$ the maximum $\ell_p$ perturbation-norm bound for the threat model against which the *classifier* has been adversarially pre-trained, the purifier is trained on a mixture of *clean* and adversarially *perturbed* examples – the latter evenly split among $\text{FGSM}_{\epsilon/2}$, $\text{PGD}_{\epsilon/2}$, $\text{FGSM}_{\epsilon}$, and $\text{PGD}_{\epsilon}$.

Additionally, care is put towards the preparation of batches each equally representative of the different types and strengths of attacks, while relying solely on dataset shuffling to benefit from regularisation due to stochastic approximation (Robbins & Monro, 1951; Bottou, 1999). Precisely, within an epoch of training, a fixed fraction of every *clean* batch resulting from the shuffled dataset is kept unprocessed – whereas the remaining portion perturbed by an even split among the attacks listed above.

Such choice is experimentally justified by the fact that any smaller subset of attack types or strength used for training of the *purifier*, or a detailedly unbalanced batch composition, always results in a worse-performing purification model. More details are provided in Appendix C.

## 4.5 INSTANCE-SPECIFIC TUNING

Further tuning of the *VAE* architecture and its usage protocol are possible – and mostly dependent on the specific data of interest.

In our investigation, we focused on image data. As a consequence, we restricted the reconstruction task to the image alone (instead of including also the conditioning tensor, as customary with *class-conditional VAE*s) and designed the decoder of the *purifier* as a Deep Generative Deconvolutional Network (DGDN) (Pu et al., 2016).

Finally, as a way to ease scalability of the method – which is proportional to the number of neurons in the *classifier* – a carefully-chosen subset of layers is used instead of the whole *internal representation*. Such *trick*, used extensively in experiments, does not significantly compromise the effectiveness of CARSO while providing tangible computational benefits. More details about principles guiding the choice of such *representation subsets* is provided in Appendix E.

Full details related to architectural choices and hyperparameters are contained in section 5 and in Appendix D.

## 5 EXPERIMENTAL ASSESSMENT

Experimental evaluation of CARSO is carried out – in terms of *robust* and *clean* image classification accuracy – within four different scenarios: *(a)*, *(b)*, *(c)* and *(d)*. The *white-box* setting is assumed throughout, as well as an $\ell_{\infty}$ norm-bound threat model. Such latter specifications describe the generally most demanding setup for adversarial defences.

## 5.1 SETUP

**Data**    In *scenarios (a)*, *(b)* and *(c)*, the CIFAR-10 (Krizhevsky, 2009) dataset is used; in *scenario (d)*, the CIFAR-100 (Krizhevsky, 2009) dataset.

**Architectures**    In *scenario (a)*, the pre-trained RESNET-18 (He et al., 2016) from Wong et al. (2020) ($1.1 \times 10^7$ parameters) is used as the *classifier*. In *scenario (b)*, the pre-trained PREACTRESNET-18 (He et al., 2016) from Rebuffi et al. (2021) ($1.25 \times 10^7$ parameters) is used instead. In *scenarios (c)* and *(d)*, the WIDERESNETs -28-10 (Zagoruyko & Komodakis, 2016) from Cui et al. (2023) ($3.65 \times 10^8$ parameters) are used, pre-trained on the respective datasets.

In all cases, the *VAE purifier* is composed of a one-layer convolutional input pre-encoder, a *FCN* context pre-encoder, a *FCN* encoder, and a deep deconvolutional decoder. Exact details on such architectures are given in Appendix D.

**Outer minimisation**    In all scenarios, *classifiers* are obtained as pre-trained models from public resources made available by the respective Authors (Wong et al., 2020; Rebuffi et al., 2021; Cui et al., 2023).

The *purifier* is trained on the *VAE* loss, using *summed pixel-wise channel-wise* binary cross entropy as reconstruction cost (for $[0, 1]$-normalised inputs). Optimisation is performed by RADAM+LOOKAHEAD (Liu et al., 2020; Zhang et al., 2019b) with *epochwise* linear one-cycle (Smith, 2017) learning rate scheduling. Scenario-specific details are provided in Appendix D.

**Inner minimisation**    In all scenarios, $\epsilon_\infty = {}^8/255$ is set as the perturbation-norm bound. In training the *purifier*, adversarial examples are obtained by maximising the categorical cross-entropy between the prediction made by the *classifier* on the clean and perturbed inputs, in a *class-untargeted* fashion.

Maximisation is performed according to the procedure described in subsection 4.4 – optimised by gradient ascent with a step size $\alpha = 0.01$ in the case of PGD. Full details and hyperparameters of the attacks are described in Appendix D.

**Evaluation**    In each scenario, we report the *clean* and *robust* test-set accuracy – the latter by means of AUTOATTACK (Croce & Hein, 2020a) – of both the *classifier* and the entire CARSO architecture.

In the case of *classifiers* alone, the *standard* version of AUTOATTACK (*AA*) is used: *i.e.* , the worst-case accuracy on a mixture of AUTOPGD attack on the cross-entropy loss (Croce & Hein, 2020a) with 100 steps, AUTOPGD on the *difference of logits ratio* loss (Croce & Hein, 2020a) with 100 steps, FAB (Croce & Hein, 2020b) with 100 steps, and the *black-box* SQUARE attack (Andriushchenko et al., 2020) with 5000 queries.

For the *end-to-end* CARSO architecture, the number of reconstructed samples per input is set to 4 in *scenarios (a)* and *(b)*, to 8 in *scenarios (c)* and *(d)*. Results are aggregated by *sum of softmaxes*, and the output class determined by its $\arg\max$. Due to the stochastic nature of the *purifier*, robust accuracy is assessed by the version of AUTOATTACK suitable for stochastic defences (*rand-AA*) – composed of AUTOPGD on the cross-entropy and *difference of logits ratio* losses, with 20 *Expectation over Transformation* (EoT) (Athalye et al., 2018b) iterations, 100 steps each.

**Computational infrastructure**    All experiments have been performed on an *NVIDIA DGX A100* system. Training and evaluation in *scenarios (a)* and *(b)* were run on 1 *NVIDIA A100* with 40 GB of dedicated memory; in *scenarios (c)* and *(d)* on all 8 of the same devices.

Training times for the purifier in all scenarios are reported in Table 1.

Table 1: Total running times for the training of the *purifier* in the different scenarios considered. *Scenarios (a)* and *(b)* employ $1\times$ whereas *scenarios (c)* and *(d)* $8\times$ GPU parallelism.

| Scenario | (a) | (b) | (c) | (d) |
|---|---|---|---|---|
| *Training time* | 129 min | 148 min | 178 min | 185 min |

## 5.2 RESULTS AND DISCUSSION

An analysis of experimental results is performed in the subsection that follows, whereas their systematic exposition is given in Table 2.

Table 2: Accuracy for the different models, datasets, and scenarios considered. **Abbreviations, col. names:** `AT/Cl`: *Clean* accuracy for the adversarially-trained *classifier*, `C+AT/Cl`: *Clean* accuracy for CARSO, `AT/AA`: *Robust* accuracy for the adversarially-trained *classifier* against standard AUTOATTACK, `C+AT/rand-AA`: *Robust* accuracy for CARSO, against stochastic-defences AUTOATTACK, `SotA AA`: *state of the art* result, per given dataset; **Abbreviations, Arch.:** `RN-18`: RESNET-18 from Wong et al. (2020), `PARN-18`: PREACTRESNET-18 from Rebuffi et al. (2021), `WRN-28-10`: WIDERESNET-28-10 from Cui et al. (2023); **Abbreviations, SotA.:** `(AT)`: result obtained by *SotA* adversarial training (CIFAR-10: Peng et al. (2023); CIFAR-100: Wang et al. (2023)), `(P)`: result obtained by *SotA* adversarial purification (CIFAR-10: Chen et al. (2023)).

| Scen. | Dataset | Arch. | AT/Cl | C+AT/Cl | AT/AA | C+AT/rand-AA | *SotA* AA |
|-------|---------|-------|-------|---------|-------|--------------|-----------|
| (a) | CIFAR-10 | RN-18 | **0.8380** | 0.7755 | 0.4336 | 0.7096 | 0.7107 (AT), **0.7324** (P) |
| (b) | CIFAR-10 | PARN-18 | **0.8353** | 0.7824 | 0.5668 | 0.6648 | 0.7107 (AT), **0.7324** (P) |
| (c) | CIFAR-10 | WRN-28-10 | **0.9216** | 0.8602 | 0.6773 | **0.7570** | 0.7107 (AT), 0.7324 (P) |
| (d) | CIFAR-100 | WRN-28-10 | **0.7385** | 0.6692 | 0.3918 | **0.6573** | 0.4267 (AT) |

***Scenarios (a) and (b)*** The adversarially pre-trained *classifier*s considered in *scenarios (a)* and *(b)* share and almost-identical architecture and provide comparable *clean* classification accuracy ($\approx 83\%$). Their *robust* accuracy is instead evidently different, consequence of the much more demanding, accuracy-focused training protocol of Rebuffi et al. (2021) compared to speed-focused Wong et al. (2020). Still far from the *SotA* for CIFAR-10 (Peng et al., 2023; Chen et al., 2023), results from *scenario (b)* are in line with the current *adversarial training* best, for a RESNET-18 on CIFAR-10, achieved by Gowal et al. (2021): $87.35\%$ *clean* and $58.63\%$ *robust* accuracy.

The adoption of CARSO strongly improves the adversarial robustness of the resulting models – attacked *end-to-end* – at the cost of decreased *clean* accuracy. Comparison in terms of robustness alone is remarkably competitive with the first positions of the ROBUSTBENCH (Croce et al., 2021) leaderboard (for the same dataset and threat model, *e.g.* Peng et al. (2023); Wang et al. (2023); Cui et al. (2023)) – even though *clean* accuracy may become a limiting factor in this regard instead. This motivates the investigation of models with a more refined pre-trained *classifier*.

***Scenario (c)*** The *classifier* considered in *scenario (c)* offers increased *clean* and *robust* accuracy *w.r.t. scenario (b)*, thanks to much more learnable parameters and a deeper structure. It currently stands in 1st position per-architecture and 4th overall in the ROBUSTBENCH leaderboard, after $> 2\times$ deeper and wider models.

The application of CARSO results – still at the cost of a *clean* accuracy penalty ($-6.14\%$) – in a significant increase ($+7.97\%$) in robust accuracy – sufficient to overtake the current best from adversarial training for CIFAR-10.

Such result also represents an improvement over the current CIFAR-10 overall best, reached by diffusion-based adversarial purification in Chen et al. (2023) ($93.16\%$ *clean* and $73.24\%$ *robust* accuracies).

***Scenario (d)*** Architecturally identical to that of *scenario (c)*, the pre-trained *classifier* used in *scenario (d)* is able to provide significant CIFAR-100 robust classification accuracy – in spite of its smaller size compared to the first model in the ROBUSTBENCH leaderboard. In terms of robust accuracy, it currently stands 1st per-architecture (WRN-28-10) and 2nd for CIFAR-100 overall.

In this case, our method produces a model able to conquer the overall *SotA* for CIFAR-100 (Wang et al., 2023) – by a large margin ($+23.06\%$). Not unlike previous cases, a *clean* accuracy toll ($-6.93\%$) is imposed by the method.

**Assessing the impact of *gradient obfuscation*** Even though the CARSO architecture is *end-to-end* algorithmically differentiable – and the integrated diagnostics included in *rand-AA* never identified the issue when attacking it *end-to-end* – we additionally guard against the eventual gradient obfuscation (Athalye et al., 2018a) induced by our method by repeating the evaluation at $\epsilon_\infty = 0.9$, verifying that below-random robust accuracy is always achieved (Carlini et al., 2019). Results of such test in all scenarios are shown in Table 3.

This concludes the evaluation of the method.

Table 3: Robust classification accuracy against AUTOATTACK, for $\epsilon = 0.9$ – as a way to assess the (lack of) impact of *gradient obfuscation* on robust accuracy evaluation.

| Scenario | (a) | (b) | (c) | (d) |
|---|---|---|---|---|
| $\epsilon = 0.9$ *acc.* | $\leq 0.0778$ | $\leq 0.0359$ | $\leq 0.0475$ | $\leq 0.0048$ |

## 5.3 LIMITATIONS AND OPEN PROBLEMS

In line with recent works aiming at the development of robustness against multiple perturbations (Dolatabadi et al., 2022; Laidlaw et al., 2021), our method imposes a decrease in *clean* accuracy *w.r.t.* the adversarially-trained *classifier* alone. In our case, such decrease is surely dependent on the use of a *VAE* as the generative purification model – result of the deliberate choice of testing the method in the worst-case for the *defender*. To overcome this limitation, more expressive and capable purification techniques (*e.g.* based on *diffusion* or *score* modelling) may be adopted. Also, in an attempt to improve cross-talk between *classifier* and *purifier* without harming the overall robustness – a similar approach to that of Liao et al. (2018) may be worth of consideration.

Finally, scalability issues may limit the applicability of CARSO – as it requires to train a *purifier* whose input is linear in the number of neurons of the *classifier*. Using a subset of layers as a surrogate for the entire *internal representation* – as described in subsection 4.5 – does mitigate the problem in practice. This, however, comes at the cost of a handcrafted, heuristic-driven selection of the most suitable representation subsets.

## 6 CONCLUSION

In this work, we presented a novel adversarial defence mechanism tightly integrating input *purification*, and classification by an adversarially-trained model – being ultimately able to improve upon current *state of the art* in CIFAR-10 and CIFAR-100 $\ell_\infty$ robust classification, both *w.r.t. adversarial training* and *purification* approaches. Such results show the value of CARSO as a viable strategy to improve adversarial robustness in visual tasks, with limited additional computational expenditure.

As a consequence, this motivates the scaling of CARSO to more challenging benchmarks and use-cases. Such adaptation would require a general re-framing of our experimental setup, and the adoption of a more capable purifier (*e.g.* based on *diffusion-* or *score*-based modelling) – able to handle increased input size and a wider variability across more classes. Furthermore, with adversarial attack generation against such generative models being an open field of research, a dedicated analysis and careful comparison of obtained results becomes necessary.

Finally, such endeavour will require a deeper understanding and the development of general, automated criteria for the selection of relevant subsets of *internal representations* – to be used in the conditioning the purification model. Candidate approaches in this regard may be rooted in the analysis of *layerwise intrinsic dimension* (Ansuini et al., 2019), or novel metrics to quantify the most informative neurons.

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

## A  ON PROJECTED GRADIENT DESCENT ADVERSARIAL TRAINING

The task of determining model parameters $\boldsymbol{\theta}^\star$ that are robust to adversarial perturbations is cast by Madry et al. (2018) as a *min-max* optimisation problem seeking to minimise *adversarial risk*, *i.e.*:

$$\boldsymbol{\theta}^\star \approx \hat{\boldsymbol{\theta}}^\star := \arg\min_{\boldsymbol{\theta}} \mathbb{E}_{(\boldsymbol{x},y)\sim\mathcal{D}} \left[ \max_{\boldsymbol{\delta}\in\mathbb{S}} \mathcal{L}\left(f\left(\boldsymbol{x}+\boldsymbol{\delta};\boldsymbol{\theta}\right),y\right) \right]$$

where $\mathcal{D}$ is the distribution over examples $\boldsymbol{x}$ and labels $y$, $f(\cdot;\boldsymbol{\theta})$ is a model with learnable parameters $\boldsymbol{\theta}$, $\mathcal{L}$ is a suitable loss function, and $\mathbb{S}$ is the set of allowed perturbations. In the case of $\ell_p$ norm-bound perturbations of maximum magnitude $\epsilon$, we can specify $\mathbb{S} := \{\boldsymbol{\delta} \mid \|\boldsymbol{\delta}\|_p \leq \epsilon\}$.

The inner optimisation problem is solved, again by Madry et al. (2018), by *Projected Gradient Descent* (PGD), an iterative algorithm whose goal is the synthesis of adversarial perturbation $\hat{\boldsymbol{\delta}} = \boldsymbol{\delta}^{(K)}$ after $K$ *gradient ascent and projection* steps defined as:

$$\boldsymbol{\delta}^{(k+1)} \leftarrow \mathfrak{P}_{\mathbb{S}}\Big(\boldsymbol{\delta}^{(k)} + \alpha \operatorname{sign}\Big(\nabla_{\boldsymbol{\delta}^{(k)}}\mathcal{L}_{ce}(f(\boldsymbol{x}+\boldsymbol{\delta}^{(k)};\boldsymbol{\theta}),y)\Big)\Big)$$

where $\boldsymbol{\delta}^{(0)}$ is randomly sampled within $\mathbb{S}$, $\alpha$ is a hyperparameter (*step size*), $\mathcal{L}_{ce}$ is the cross-entropy function, and $\mathfrak{P}_{\mathbb{A}}$ is the Euclidean projection operator onto set $\mathbb{A}$, *i.e.* $\mathfrak{P}_{\mathbb{A}}(\boldsymbol{a}) := \arg\min_{\boldsymbol{a}'\in\mathbb{A}} \|\boldsymbol{a} - \boldsymbol{a}'\|_2$ .

The outer optimisation is carried out by simply training $f(\cdot;\boldsymbol{\theta})$ on the examples found by PGD against current model parameters – and their pre-perturbation labels. The overall procedure just described constitutes PGD *adversarial training*.

## B  ON THE FUNCTIONING OF (CONDITIONAL) VARIATIONAL AUTOENCODERS

Variational autoencoders (*VAE*s) (Kingma & Welling, 2014; Rezende et al., 2014) learn, from data, a generative distribution of the form $p(\boldsymbol{x},\boldsymbol{z}) = p(\boldsymbol{x}\,|\,\boldsymbol{z})p(\boldsymbol{z})$, where probability density $p(\boldsymbol{z})$ represents a prior over latent variable $\boldsymbol{z}$, and $p(\boldsymbol{x}\,|\,\boldsymbol{z})$ is the likelihood function, which can be used to sample data of interest $\boldsymbol{x}$, given $\boldsymbol{z}$.

Training is carried out by maximising a variational lower bound $-\mathcal{L}_{\text{VAE}}(\boldsymbol{x})$ on the log-likelihood $\log p(\boldsymbol{x})$ – which is a proxy for the *Evidence Lower Bound* (*ELBO*) – *i.e.*:

$$-\mathcal{L}_{\text{VAE}}(\boldsymbol{x}) := \mathbb{E}_{q(\boldsymbol{z}\,|\,\boldsymbol{x})}[\log p(\boldsymbol{x}\,|\,\boldsymbol{z})] - \text{KL}(q(\boldsymbol{z}\,|\,\boldsymbol{x})\|p(\boldsymbol{z}))$$

where $q(\boldsymbol{z}\,|\,\boldsymbol{x}) \approx p(\boldsymbol{z}\,|\,\boldsymbol{x})$ is an approximate posterior and $\text{KL}(\cdot\|\cdot)$ is the Kullback-Leibler divergence.

By parameterising likelihood with a *decoder ANN* $p_{\boldsymbol{\theta}_{\text{D}}}(\boldsymbol{x}\,|\,\boldsymbol{z};\boldsymbol{\theta}_{\text{D}}) \approx p(\boldsymbol{x}\,|\,\boldsymbol{z})$, and a possible variational posterior with an *encoder ANN* $q_{\boldsymbol{\theta}_{\text{E}}}(\boldsymbol{z}\,|\,\boldsymbol{x};\boldsymbol{\theta}_{\text{E}}) \approx q(\boldsymbol{z}\,|\,\boldsymbol{x})$, the parameters $\boldsymbol{\theta}_{\text{D}}^\star$ of the generative model better reproducing the data may be learned – jointly with $\boldsymbol{\theta}_{\text{E}}^\star$ – as:

$$\begin{aligned}
\boldsymbol{\theta}_{\text{E}}^\star, \boldsymbol{\theta}_{\text{D}}^\star := \quad & \\
\arg\min_{(\boldsymbol{\theta}_{\text{E}},\boldsymbol{\theta}_{\text{D}})} & \mathcal{L}_{\text{VAE}}(\boldsymbol{x}) = \\
\arg\min_{(\boldsymbol{\theta}_{\text{E}},\boldsymbol{\theta}_{\text{D}})} & \mathbb{E}_{\boldsymbol{x}\sim\mathcal{D}} \left[ -\mathbb{E}_{\boldsymbol{z}\sim q_{\boldsymbol{\theta}_{\text{E}}}(\boldsymbol{z}\,|\,\boldsymbol{x};\boldsymbol{\theta}_{\text{E}})} \left[\log p_{\boldsymbol{\theta}_{\text{D}}}(\boldsymbol{x}\,|\,\boldsymbol{z};\boldsymbol{\theta}_{\text{D}})\right] + \text{KL}(q_{\boldsymbol{\theta}_{\text{E}}}(\boldsymbol{z}\,|\,\boldsymbol{x};\boldsymbol{\theta}_{\text{E}})\|p(\boldsymbol{z})) \right]
\end{aligned}$$

where $\mathcal{D}$ is the distribution over (training) examples $\boldsymbol{x}$.

From the practical viewpoint, optimisation relies on the empirical evaluation of $\mathcal{L}_{\text{VAE}}(\boldsymbol{x};\boldsymbol{\theta})$ on minibatches of data, with $-\mathbb{E}_{\boldsymbol{z}\sim q_{\boldsymbol{\theta}_{\text{E}}}(\boldsymbol{z}\,|\,\boldsymbol{x};\boldsymbol{\theta}_{\text{E}})} \left[\log p_{\boldsymbol{\theta}_{\text{D}}}(\boldsymbol{x}\,|\,\boldsymbol{z};\boldsymbol{\theta}_{\text{D}})\right]$ replaced by a *reconstruction cost*

$$\mathcal{L}_{\text{Reco}}(\boldsymbol{x},\boldsymbol{x}') \geq 0 \,|\, \mathcal{L}_{\text{Reco}}(\boldsymbol{x},\boldsymbol{x}') = 0 \iff x = x' \,.$$

Generation of new data according to the fitted model is achieved by sampling from

$$p_{\boldsymbol{\theta}_{\mathrm{D}}^{\star}}(\boldsymbol{x} \mid \boldsymbol{z}; \boldsymbol{\theta}_{\mathrm{D}}^{\star})\Big|_{\boldsymbol{z} \sim p(\boldsymbol{z})}$$

*i.e.* decoding samples from $p(\boldsymbol{z})$.

The setting is analogous in the case of *conditional* Variational Autoencoders (Sohn et al., 2015; Yan et al., 2016) (see section 3), where (conditional) sampling is achieved by

$$\boldsymbol{x}_{\boldsymbol{c}_j} \sim p_{\boldsymbol{\theta}_{\mathrm{D}}^{\star}}(\boldsymbol{x} \mid \boldsymbol{z}, \boldsymbol{c}; \boldsymbol{\theta}_{\mathrm{D}}^{\star})\Big|_{\boldsymbol{z} \sim p(\boldsymbol{z}); \ \boldsymbol{c}=\boldsymbol{c}_j} .$$

## C  JUSTIFICATION OF ADVERSARIALLY-BALANCED BATCHES

During the incipient phases of experimentation, preliminary tests were performed on the MNIST (LeCun & Cortes, 2010) and Fashion-MNIST (Xiao et al., 2017) datasets – using a conditional *VAE* as *purifier*, and small *fully connected* (FCN) or *convolutional ANN*s as *classifiers*. Adversarial examples were generated against the (adversarially) pre-trained *classifier*, and tentatively denoised by the *purifier* – sampling only once from the distribution of reconstructions. The resulting recovered inputs were classified again by the *classifier* and the difference in overall accuracy recorded.

Importantly, such tests were not meant to assess the *end-to-end* adversarial robustness of the whole architecture, but just to tune the training protocol for the *purifier*.

If the purification machinery had been trained with PGD-generated adversarial examples only – the *de facto* standard for adversarial training of classifiers:

- Unsatisfactory *clean* accuracy was reached upon convergence, speculatively a consequence of the *VAE* never having been trained on *clean*-to-*clean* example mapping;
- Persistent vulnerability to same norm-bound FGSM perturbations was noticed;
- Persistent vulnerability to smaller norm-bound FGSM and PGD perturbations was noticed.

In order to mitigate such issues, the composition of adversarial examples used for training the *purifier* was adapted to specifically counter each of the pitfalls identified. Adoption of any smaller subset *w.r.t.* that described in subsection 4.4 resulted in unsatisfactory accuracy *w.r.t.* at least one of the (sub)cases listed above.

At that point, another problem emerged: if the *adapted* adversarial training protocol had been carried out by producing homogeneous batches from the same type/strength of attack (or clean), the resulting test accuracy across the remaining cases varied significantly as a function of previous batch ordering.

Such observation lead to the final formulation of the training protocol – with detailedly *balanced* batches (see subsection 4.4) – which mitigates successfully all the issues described so far.

## D  ARCHITECTURAL DETAILS AND HYPERPARAMETERS

In the appendix that follows, we provide exact details about the architectures (subsection D.1) and hyperparameters (subsection D.2) used in the experimental phase of our work.

### D.1  ARCHITECTURES

Here, we describe the specific architectural choices for the individual parts of the *purifier*.

The *input pre-encoder* is always composed of one *biased* convolutional layer with `kernel size=` `4`, `stride= 1` and `padding= 0`. The number of output channels is the least – compatible with given *kernel size*, *stride*, and *padding* – s.t. the number of scalars necessary to represent the *pre-encoded input* is greater or equal to that of *input* itself. *Implementation-wise*, such number is

computed programmatically and never set explicitly. This choice never results in an actual *input compression*.

Architectures used for the *representation pre-encoder* and encoder are shown in Table 4. The subsets of layers used, on a case-by-case basis, instead of the whole *internal representation* of the *classifier*, are reported in subsection D.2, additional hyperparameters left unspecified are provided in Table 7.

Table 4: Architectures for *representation pre-encoder* and *encoder* of the *purifier*. Symbols $f_i$ and $f_o$ denote respectively the number of input features in, and output features of, the given (pre-)encoder. Specific values for different scenarios, datasets, and architectures are reported in Table 7.

| *Repr. pre-encoder* | Encoder |
|---|---|
| `Linear(features_in=`$f_i$`, features_out=`$2 \times f_o$`)` | `Linear(features_in=`$f_i$`, features_out=`$f_i + f_o/2$`)` |
| `Batch Normalisation` | `Batch Normalisation` |
| `LeakyReLU(slope=0.01)` | `LeakyReLU(slope=0.01)` |
| `Linear(features_in=`$2 \times f_o$`, features_out=`$f_o$`)` | `Linear(features_in=`$f_i + f_o/2$`, features_out=`$f_o$`)` |
| `Batch Normalisation` | `Batch Normalisation` |
| `LeakyReLU(slope=0.01)` | `LeakyReLU(slope=0.01)` |
| `Sigmoid` | `Tanh` |

The sampler used for the generation of latent variables $z$ during training is a reparameterised (Kingma & Welling, 2014) Normal sampler $z \sim \mathcal{N}(\mu_e, \sigma_e)$ whose characteristic parameters $\mu_e$ and $\sigma_e$ are the output of two independent `Linear` layers receiving as input the output of the encoder. The output size of the sampler is provided in Table 7. During inference, $z$ is sampled (again by reparameterisation) from the *i.i.d* Standard Normal $z \sim \mathcal{N}(0, 1)$ (*i.e.* from its prior).

*Decoder* architectures are shown in Table 5, according to the specific data to be generated. The number of output channels $c_o$ is the same as in the original data (*i.e.* 3 for the data considered). The number of input channels $c_i$ is the `Sample Size` provided in Table 7.

Table 5: Architecture for the decoder of the *purifier*. The name `TrConv` denotes transposed convolutions. In the description of deconvolutional layers, the following shorthand notation is used: `ch_in`: number of input channels, `ch_out`: number of output channels, `k`: square kernel size, `s`: isotropic stride, `p`: isotropic padding. Deconvolutional layers are *bias-free*. Symbols $c_i$ and $c_o$ denote respectively the number of input channels in, and output channels of, the decoder.

| |
|---|
| `TrConv2D(ch_in=`$c_i$`, ch_out=`$c_i/2$`, k=4, s=1, p=0)` |
| `Batch Normalisation` |
| `LeakyReLU(slope=0.01)` |
| `TrConv2D(ch_in=`$c_i/2$`, ch_out=`$c_i/4$`, k=4, s=2, p=1)` |
| `Batch Normalisation` |
| `LeakyReLU(slope=0.01)` |
| `TrConv2D(ch_in=`$c_i/4$`, ch_out=`$c_i/8$`, k=4, s=2, p=1)` |
| `Batch Normalisation` |
| `LeakyReLU(slope=0.01)` |
| `TrConv2D(ch_in=`$c_i/8$`, ch_out=`$c_o$`, k=4, s=2, p=1)` |
| `Batch Normalisation` |
| `Sigmoid` |

## D.2 HYPERPARAMETERS

Here, we report the hyperparameters used for *adversarial example generation* and *optimisation* during the training of the *purifier*, and those related to *purifier* model architectures.

**Attacks**  Hyperparameters for the specific adversarial attacks employed are shown in Table 6. The value of $\epsilon_\infty$ is chosen according to the specific scenario and use of the attack. With the only exception of $\epsilon_\infty$, AUTOATTACK is to be considered *hyperparameter-free*.

Table 6: Hyperparameters for the attacks used for training the *purifier* adversarially. The entry CCE denotes the *Categorical CrossEntropy* loss function. The $\ell_\infty$ threat model is assumed, and all inputs are linearly rescaled within $[0.0, 1.0]$ before the attack.

|  | FGSM | PGD |
|---|---|---|
| Input clipping | $[0.0, 1.0]$ | $[0.0, 1.0]$ |
| # of steps | 1 | 40 |
| Step size | $\epsilon_\infty$ | 0.01 |
| Loss function | *CCE* | *CCE* |
| Optimiser |  | *SGD* |

**Architectures**  Table 8 shows the subset of layers used as a surrogate of the entire *internal representation* of the *classifier* (see subsection 4.5). Names of the layers refer to those used in the corresponding *classifier* implementation.

Table 7: Architectural hyperparameters for the *representation pre-encoder* and *actual* encoder used within CARSO. Output size for the $z$ sampler is also reported. *Implementation-wise*, input sizes for the encoder are never set explicitly, but computed from the concatenation of *pre-encoded representation* and *pre-encoded input*.

|  | *Repr. pre-enc.* $f_i$ | *Repr. pre-enc.* $f_o$ | Enc. $f_i$ | Enc. $f_o$ | Sample size |
|---|---|---|---|---|---|
| RESNET-18 (Wong et al., 2020) | $2.04810 \times 10^5$ | 512 | (computed) | 192 | 128 |
| PREACTRESNET-18 (Rebuffi et al., 2021) | $2.04810 \times 10^5$ | 512 | (computed) | 192 | 128 |
| WRN-28-10 (Cui et al., 2023) (CIFAR-10) | $5.73450 \times 10^5$ | 512 | (computed) | 192 | 128 |
| WRN-28-10 (Cui et al., 2023) (CIFAR-100) | $2.86820 \times 10^5$ | 2816 | (computed) | 192 | 128 |

Table 8: *Classifier* layer names used as a subset of the *internal representation* fed to the *representation pre-encoder* of the *purifier*. The names in *scenarios (a)* and *(b)*, though different, encode the same layers along (essentially) the same architecture.

| Scen. (a) | Scen. (b) | Scen. (c) | Scen. (d) |
|---|---|---|---|
| model.layer1.1.conv2 | layer_0.1.conv_2d_2 | layer.0.block.1.conv_1 | layer.1.block.0.conv_0 |
| model.layer2.0.conv2 | layer_1.0.conv_2d_2 | layer.1.block.0.shortcut | layer.1.block.1.conv_1 |
| model.layer2.0.shortcut.0 | layer_1.0.shortcut | layer.1.block.1.conv_1 | layer.2.block.0.conv_1 |
| model.layer3.0.conv2 | layer_2.0.conv_2d_2 | layer.1.block.2.conv_1 | layer.2.block.1.conv_1 |
| model.layer3.0.shortcut.0 | layer_2.0.shortcut | layer.2.block.0.shortcut | layer.2.block.2.conv_1 |
| model.layer3.1.conv2 | layer_2.1.conv_2d_2 | layer.2.block.1.conv_1 | logits |
| model.layer4.0.conv2 | layer_3.0.conv_2d_2 | layer.2.block.2.conv_1 |  |
| model.layer4.0.shortcut.0 | layer_3.0.shortcut | layer.2.block.3.conv_1 |  |
| model.layer4.1.conv2 | layer_3.1.conv_2d_2 | logits |  |
| model.linear | logits |  |  |

**Training**  Table 9 collects hyperparameters governing the training of the *purifier* in the different scenarios considered.

Table 9: Hyperparameters for the training of *purifiers*, grouped by scenario. The entry `CCE` denotes the *Categorical CrossEntropy* loss function. The *epochwise* specification of the *LR* scheduler refers to the fact it is applied after each epoch (as opposed to *batchwise*, as more common in practice).

|  | All *scenarios* | *Sc. (a)* | *Sc. (b)* | *Sc. (c)* | *Sc. (d)* |
|---|---|---|---|---|---|
| Optimiser | RADAM+LOOKAHEAD | | | | |
| RADAM $\beta_1$ | 0.9 | | | | |
| RADAM $\epsilon$ | $10^{-8}$ | | | | |
| RADAM *Weight Decay* | None | | | | |
| LOOKAHEAD *averaging decay* | 0.8 | | | | |
| Loss function | *CCE* | | | | |
| *LR* Scheduling | linear, 1-cycle, *epochwise* | | | | |
| RADAM $\beta_2$ | | 0.999 | 0.999 | 0.999 | 0.99 |
| LOOKAHEAD steps | | 5 | 5 | 6 | 6 |
| Epochs | | 150 | 150 | 150 | 200 |
| Increasing *LR* epochs | | 50 | 50 | 37 | 50 |
| Decreasing *LR* epochs | | 100 | 100 | 113 | 150 |
| Minimum *LR* | | $5 \times 10^{-8}$ | $5 \times 10^{-8}$ | $5 \times 10^{-9}$ | $5 \times 10^{-9}$ |
| Maximum *LR* | | 0.05376 | 0.05376 | 0.08 | 0.065 |
| Batch size | | 1536 | 1536 | 800 | 128 |
| Adversarial *batch fraction* | | 0.4 | 0.4 | 0.4 | 0.12 |
| Sampled reconstructions per input | | 4 | 4 | 8 | 8 |

## E    ON THE CHOICE OF REPRESENTATION SUBSETS

In selecting the subsets of *internal representations* described in subsection 4.5, we limit ourselves to groups of entire layers – though other options are technically possible.

To balance model size, informativeness, and robustness, layers are picked with approximately-even spacing along model depth – avoiding in such way excessive mutual correlation. Given the decreasing single-layer size from input to output across the architecture, layers from the first RESNET blocks are preferentially avoided. However, especially in *scenarios (c)* and *(d)* – where the overall increased layer width requires a more careful choice to keep subset size controlled – the importance of including at least one layer from such blocks becomes more evident.

The subsets finally chosen, and shown in subsection D.2, are the result of those heuristically-motivated guidelines – complemented with some experimentation.

