# OpenReview forum: "CARSO: Blending Adversarial Training and Purification Improves Adversarial Robustness"
_ICLR.cc/2024/Conference — Submitted to ICLR 2024_

### Official Review · Reviewer_VTuG · 2023-10-26

**Soundness:** 2 fair
**Presentation:** 2 fair
**Contribution:** 2 fair
**Rating:** 1
**Confidence:** 5

**Summary:**

In this work, the authors introduce a novel adversarial defense mechanism named CARSO, which combines adversarial training and adversarial purification techniques to enhance the robustness of image classification models. CARSO leverages an adversarially-trained classifier to map potentially perturbed input data to a distribution of clean reconstructions, aggregating multiple samples from this distribution to make robust predictions. Experimental results across various benchmarks demonstrate that CARSO effectively defends against a wide range of adaptive attacks, significantly outperforming existing methods in terms of robust classification accuracy on datasets like CIFAR-10 and CIFAR-100, especially against AutoAttack.

**Strengths:**

Authors proposed a new method combining AT and purification methods for advesarial attack defence.

**Weaknesses:**

1. Author used a method of mixing purification and AT,
 but I don't think it is fair to only have an ablation study in AT.
I wonder comparison between purification method vs CARSO + purification method.

2. It was said that the motivation for separating the scenarios was due to a lack of clean image accuracy,
but as a result, the same difference is shown in (c) and (d).
 So, I don’t think that dividing the scenario and showing the experiment is an important part of the paper.
Rather, the logic that it was used in the expect of obtaining an internal representation
from a more refined classifier in order to perform SOTA on a robust image seems more appropriate.

3. Author made an analogy about the method using terms(cued recall and recognition) from cognitive science,
but it doesn't seem to be clear.

4. Too much limited and insufficient experiments: There are no state-of-the-art defense baselines such as AWP [1], SCORE [2], and ADML [3], and no larger-scale dataset such as ImageNet. In addition, based on ADML, not only CNN structure and Transformer structures seems needed to validate.

5. Table 2 conducted an ablation study comparing effect of the CARSO method. Clean image performance significantly decreases but the current state-of-the-art ADML method highly increases clean image performacne despite few epochs within 3-5 epochs based on their official code. I did not understand the major contribution compared by ADML.


---

**References**

[1] Wu, Dongxian, Shu-Tao Xia, and Yisen Wang. "Adversarial weight perturbation helps robust generalization." Advances in Neural Information Processing Systems 33 (2020): 2958-2969.

[2] Pang, Tianyu, et al. "Robustness and accuracy could be reconcilable by (proper) definition." International Conference on Machine Learning. PMLR, 2022.

[3] Lee, Byung-Kwan, Junho Kim, and Yong Man Ro. "Mitigating adversarial vulnerability through causal parameter estimation by adversarial double machine learning." Proceedings of the IEEE/CVF International Conference on Computer Vision. 2023.

---

**Post Rebuttal**

Although the authors performed rebuttal, the reviewer would like to strongly recommend this paper be modified due to the following reasons:

(1) It is not convincing that adversarial purification model can improve robustness in current version of this paper. For convincing argument, it is much greater to change the motivation of this paper to ***adversarial purification improves robustness*** in introduction, and prove it in the following sections with theoretical or empirical background. In other words, the authors should show that naively denoising images are significantly different than adversarial purification, and adversarial purification is really the most important part in general adversarial robustness.

(2) More importantly, this paper should make the reader think of its paper as expandable work to general adversarial robustness. For example, it is better to validate that the performance of "CARSO" + "MART" or "AWP" or "SCORE" or "ADML" is stronger than that of "CARSO" + "AT(baseline)". This is because CARSO is adversarial purification model.

(3) There should exist a experiment of considering state-of-the-art image denoising model based on deep learning. Simply, the reviewer call this model as "A". Here, the authors should validate that "CARSO" + "MART" or "AWP" or "SCORE" or "ADML" is much stronger than adversarially training "A" + "MART" or "AWP" or "SCORE" or "ADML", where "A" should be, of course, trained with same adversarial strategy with CARSO. This is because CARSO is, again, adversarial purification model.

**Questions:**

Refer to Weaknesses

---

> ### Author Response · Authors · 2023-11-16
> **Rebuttal to Official Review by Reviewer VTuG (part 1 of 4)**
>
> We thank the Reviewer for his/her comments and suggestions. However, we cannot avoid noticing that parts of the review are based on either a misinterpretation of the contents of our manuscript, or unfair representations of existing published works.
>
> 1.   **On ablation studies against AT- and purification-based methods**
>      The Reviewer is correct in stating that our proposal is based on a mix of *adversarial training* and *adversarial purification*. However, comparisons contained in *Table 2* and commented in *Section 5* are not to be intended as proper ablation studies: they assess the clean and robust accuracy of CARSO against the **sole adversarially-trained classifier model** used as part of CARSO itself, as a baseline.
>      In any case, they may also be conceptually seen as ablation studies where the purification part is removed, and we assume the reviewer refers to such fact. The purification part of CARSO – however – is by definition **conditional on the representation** of an existing adversarially-trained classifier. As a consequence, a proper ablation study requiring that only the purification portion of CARSO is used, would be ill-defined.
>      In general, one may use known results from published literature to virtually *ablate away* the adversarial training contribution to CARSO, by comparing our method directly against purification-only adversarial defences such as [1] from 2015 (where, exactly as in CARSO, VAEs are used as purifiers) or [2] (which is the overall *state of the art* in such respect).
>      In the first case, VAEs have even been shown to tragically **decrease** the robustness of the resulting system – requiring the use of much more modern and capable *diffusion-* or *score-*based models to obtain a self-standing purification-based defence. Such information is contained in the *Introduction* of the paper, and in *Section 2*, *Purification as a defence*.
>
>      In the second case, such comparison is directly contained in *Table 2*, columns 7 and 8, and constitutes a main element of the experimental assessment of our method against existing, published ones.
>      Finally, we were unfortunately unable to understand the suggestion of a *“purification method vs CARSO + purification method”* comparison, and would kindly ask the Reviewer to clarify.
>
> 2.   **On the separation of scenarios and its consequences**
>      The Reviewer states that *“the motivation for separating the scenarios was due to a lack of clean image accuracy”*. However, no such statement is contained in any part of the paper. The only principle followed in determining the different scenarios was the choice of **dataset and *classifier* model used** as a starting point for CARSO. For completeness, a recap of such differences is shown below:
>
>      -   *Scenario (a)*: CIFAR-10 / ResNet-18 from [3];
>
>      -   *Scenario (b)*: CIFAR-10 / PreAct ResNet-18 from [4];
>
>      -   *Scenario (c)*: CIFAR-10 / Wide ResNet-28-10 from [5];
>
>      -   *Scenario (d)*: CIFAR-100 / Wide ResNet-28-10 from [5];
>
> Unfortunately, we were not able to understand the next part of the question, which in any case seems to follow logically from the initial statement, not contained in the paper. We would kindly ask the Reviewer a clarification in such regard.
>
> 3.   **On the neurocognitive inspiration of CARSO**
>
>      In the *Introduction* of our paper, we state that our method has been inspired by the *neurocognitive processes underlying cued recall and recognition*. We would like to reiterate that such consideration was neither meant to be an analogical explanation, nor that our method is endowed by some kind of neurobiological plausibility: the works mentioned right after (*i.e.* [9], [10] and [11]) provided a suggestion towards the development of our main contribution: the *representation-conditional* generation of (purified, in our case) images.
>      Similarly to how a biological stimulus (of same or different modality) can evoke a neuronal ensemble whose projection onto another ensemble may directly influence a collateral task – the whole process of image purification and subsequent classification (cued recognition) within CARSO is conditioned by the evoked internal representation within the classifier, when it is fed the original, potentially perturbed, input (cue). As part of such process, additionally, the actual classification does not occur directly on the original input (cue), but on a *version* of it that is *recalled* from the results of *experience* (the training of the VAE), idealised in its form (in our case: purified from perturbations) – *i.e.* constituting a *cued recall* of the original.

---

> ### Author Response · Authors · 2023-11-16
> **Rebuttal to Official Review by Reviewer VTuG (part 2 of 4)**
>
> 4.    **On the the limits to our experimental assessment**
>
> We do agree with the Reviewer that extending the experimental assessment of our method to other datasets beyond CIFAR-10/100 (*e.g.* scaling up to ImageNet) would constitute a valuable extension to out work – and we are actively working towards such end.
> However, we would like to restate once more that our work is fundamentally concerned with whether it is possible to blend *adversarial training* and *adversarial purification* by conditioning the purification process on the internal representation of the classifier. With no prior work following such line of thought – before trying to *scale up* – we have been careful to ensure significant conclusions may be reached from a relatively simple, *proof-of-concept* set of experiments.
> Indeed, we deliberately choose a VAE as a purification model – whose use as an adversarial purifier has been long abandoned due to its relatively scarce expressiveness in image reconstruction – to allow for exact differentiability of the sampling process, and to emphasise that it is the **interplay with the AT-trained classifier** that is able to strongly improve robustness; and not the exploitation of an already *SotA* purification approach. In such constrained setting, and with no significant risk of robustness overestimation (as would instead happen when gradient approximations are used [2]) we still manage to overcome both state-of-the-art *AT-based* models and diffusion-based *purification* methods.
> A *scale-up* of our approach is indeed possible – both in terms of data size and task complexity. Reasonably doing so would require the use of a more modern and capable purification architecture (*e.g.* *score-*, *diffusion-* or *flow-* based), the complete rework of the experimental assessment (since adversarial attacks against such generative model are an active field of research – and doubts have been cast around the adequacy of AutoAttack in such sense), and significant engineering contributions. We defend that including such more complex scenario in our work, at this stage, would damage clarity of our contribution.
>
> 4 bis. **On comparison with additional *SotA* baselines**
> With respect to explicit comparisons, within the main body of the paper, to existing adversarial defence methods, we limited ourselves to two categories of approaches: those (*AT-based*) used as *classifier*s as part of the CARSO architecture, and those (both *AT-* and *purification-*based, called by us *state of the art*, with this precise definition) able to obtain – for each dataset – the highest $\ell_{\infty}$ robust accuracy against AutoAttack at the moment of writing (for future reference: September 2023), according to the contents of the original paper. None of the methods mentioned by the Reviewer satisfied such criteria for inclusion.
> In any case, a comparison can be made on the spot:
>
> -   <u>AWP</u>: The best-performing method (RST-AWP WRN-28-10) against $\ell_{\infty}$ AutoAttack contained in Tables 1 and 2 of [6] for CIFAR-10 is able to reach $60.05\%$ robust accuracy. The best performing for CIFAR-100 (AT-AWP WRN-34-10) is able to reach $30.71\%$.
>     This can be compared to the results of CARSO (using the WRN-28-10 classifier from [5]): $75.70\%$ for CIFAR-10 and $65.73\%$ for CIFAR-100, which constitute a very strong improvement.
> -   <u>SCORE</u>: Similarly, the best performing model ($\beta=4$ WRN-28-10) according to *Table 3* of [7], for CIFAR-10, reaches $61.66\%$ robust accuracy. The best for CIFAR-100 ($\beta=4$ WRN-28-10) reaches $ 31.21\%$. Results for CARSO (using the WRN-28-10 classifier from [5]) are: $75.70\%$ for CIFAR-10 and $65.73\%$ for CIFAR-100, which constitute again a very strong improvement.
> -   <u>ADML:</u> According to *Table 1* and *Table 2* of the paper [8], regardless of the architecture used for the model, the best-performing model on CIFAR-10 (WRN-70-10, more than twice as deep *w.r.t.* our base *classifier*!) obtains $63.1\%$ robust accuracy. The same model, for CIFAR-100, reaches $37.6\%$. Again, compared to results for CARSO (using the WRN-28-10 classifier from [5]) – $75.70\%$ for CIFAR-10 and $65.73\%$ for CIFAR-100 – our method determines a very strong improvement.
>     Additionally, we should remind the Reviewer that ADML is covered by the *concurrent work* clause for the ICLR conference (publishing date of the method: Jun 2023).
>
> Significantly, for all methods mentioned above, not only CARSO – but even our baselines from [5] – were able to clearly improve upon respective best published results (nominally: $67.73\%$ for CIFAR-10 and $39.18\%$ for CIFAR-100 for the best of such baselines), by using equal or (much) smaller architectures. Direct comparisons of CARSO to such baselines are shown in *Table 2* of our work.

---

> ### Author Response · Authors · 2023-11-16
> **Rebuttal to Official Review by Reviewer VTuG (part 3 of 4)**
>
> 5.    **On comparison with *ADML* and clean accuracy**
>
> As mentioned in our answer to (1), the contents of *Table 2* have not to be considered as an ablation study (and in fact, such designation is absent from our paper). We refer to that specific answer for more details.
> Additionally, the Reviewer further points out two different elements of discussion:
>
> -   **On clean accuracy**
>     As the Reviewer correctly notices, the application of CARSO – in comparison to the sole *classifier* baseline – does impose a marked *clean* accuracy toll. We are aware of such downside – and we clearly state it within the text of the paper. A large part of such effect has to be imputed to the very simple architecture chosen (deliberately) for the purifier – a variational autoencoder – whose limited ability in accurate image generation is known. Such choice was motivated (as we state in *Subsection 5.3*) by the need to both test our element of novelty (*representation-conditional purification*) in a clearly unfavourable scenario for the defender, from the *purification* viewpoint (otherwise the results could have been attributed to the strength of the purifier alone) and to ensure exact differentiability of the purification machinery – lest to avoid the pitfalls identified by [2] in the evaluation of purification-based adversarial defences with approximated gradients. In any case, we remind the reviewer that ADML is covered by the *concurrent work* clause for the ICLR conference (publishing date of the method: Jun 2023).
>     On a side note, we are always glad to see novel research lines in the field – targeting the problem of *clean* accuracy decrease as a result of robustness increases. In fact, such results – in turn – mitigate the clean accuracy downsides of a method such as CARSO, if they are used as *base* classifiers.
>
> -   **On the comparison with ADML**
>     The Reviewer states *“I did not understand the major contribution compared by ADML.”*. We would start by reminding the reviewer (once more) that ADML is covered by the *concurrent work* clause for the ICLR conference (publishing date of the method: Jun 2023). In any case, we will delve into the merit of the question in the following.
>
>     Despite being both contributions to the field of adversarially-robust DL – CARSO and ADML are substantially different approaches from the methodological viewpoint. ADML represents (to the best of our understanding) a strong, general-purpose method to boost the effectiveness of adversarial training, by leveraging the principles of *double machine learning* **within AT itself**. On the other hand, CARSO stemmed from an investigation around the question: *Is there a way to organically blend adversarial training and purification-based defences?* – and as such the paper offers a smaller-scale, proof-of-concept experimental assessment showing that **even under some very stringent constraints of exact differentiability and limited expressiveness** of the purifier, *SotA* results can be reached in terms of CIFAR-10 and CIFAR-100 $\ell_{\infty}$ AutoAttack robust accuracy.
>     Given the completely different research questions the two methods try to answer, such justification alone would suffice to affirm the dignity of the contribution brought upon by CARSO.
>     But, even beyond such consideration, the best results reported in the respective papers (specifically referred to as an answer to (4)), show that CARSO (building upon a WRN-28-10; more than half the depth of the best ADML counterpart!) is able to significantly improve upon the  $\ell_{\infty}$ AutoAttack robust accuracies for CIFAR-10 and CIFAR-100 (the datasets we tested so far). In the case of CIFAR-100, particularly, **an almost twice improvement is shown** *w.r.t.* ADML. We specifically refer to the  answer to (4) for precise quantitative comparisons.

---

> ### Author Response · Authors · 2023-11-16
> **Rebuttal to Official Review by Reviewer VTuG (part 4 of 4)**
>
> ---
>
> ### References
>
> [1] Shixiang Gu and Luca Rigazio. Towards deep neural network architectures robust to adversarial examples. In Workshop Track of the International Conference on Learning Representations, 2015.
> [2] Huanran Chen, Yinpeng Dong, Zhengyi Wang, Xiao Yang, Chengqi Duan, Hang Su, and Jun Zhu. Robust classification via a single diffusion model, 2023.
> [3] Eric Wong, Leslie Rice, and J. Zico Kolter. Fast is better than free: Revisiting adversarial training. In International Conference on Learning Representations, 2020.
> [4] Sylvestre-Alvise Rebuffi, Sven Gowal, Dan Andrei Calian, Florian Stimberg, Olivia Wiles, and Timothy Mann. Data augmentation can improve robustness. In Advances in Neural Information Processing Systems, 2021.
> [5] Jiequan Cui, Zhuotao Tian, Zhisheng Zhong, Xiaojuan Qi, Bei Yu, and Hanwang Zhang. Decoupled Kullback-Leibler divergence loss, 2023.
> [6] Wu, Dongxian, Shu-Tao Xia, and Yisen Wang. "Adversarial weight perturbation helps robust generalization." Advances in Neural Information Processing Systems 33 (2020): 2958-2969.
> [7] Pang, Tianyu, et al. "Robustness and accuracy could be reconcilable by (proper) definition." International Conference on Machine Learning. PMLR, 2022.
> [8] Lee, Byung-Kwan, Junho Kim, and Yong Man Ro. "Mitigating adversarial vulnerability through causal parameter estimation by adversarial double machine learning." Proceedings of the IEEE/CVF International Conference on Computer Vision. 2023.
> [9] Endel Tulving and Donald M. Thomson. Encoding specificity and retrieval processes in episodic memory. In Psychological Review, 1973.
> [10] James Eric Eich. The cue-dependent nature of state-dependent retrieval. In Memory & Cognition, 1980.
> [11] J.J. Medina. The biology of recognition memory. In Psychiatric Times, 2008.

---

> ### Author Response · Authors · 2023-11-23
> **Comment to the post-rebuttal review of Reviewer VTuG (part 1 of 2)**
>
> We thank the *Reviewer* for his/her *post-rebuttal* comment. We also apologise for the initial confusion caused by the lack of notification about the publishing of such comment – having it been written as an edit to an already-published review instead of a new comment, necessary to ensure a notification to the *Authors*.
>
> As far as points of merit are concerned, we would start by clarifying once more that CARSO is **not *just* a purification model**  – at least not in the conventional sense. Indeed, CARSO is an adversarial defence composed of **two, different models** (the *classifier* and the *purifier*, the former of which can be obtained as a pre-trained model) operating in close synergy. Specifically, **the purifier performs purification from the *internal representation* of the input produced by the *classifier***, and not directly from the input. Afterwards, **the same *classifier* performs classification on many samples obtained from the *purifier***. This specific principle of operation allows to defend against the pitfalls of *naïve* adversarial purification outlined in [1], and allows to exploit the specific properties of gradient arising at the level of the neurons in the *classifier* during an adversarial attack – as outlined in *Subsection 4.2*.
>
> We will address each of the points raised, more in detail, in the following.
>
> 1.   We understand that the *Reviewer* would prefer to see – instead of one we submitted – a paper showing clearly that adversarial purification alone is able to contribute significantly to the development of general adversarial robustness. Additionally, he/she would like evidence that adversarial purification is different from *naïve* denoising of the input.
>      While we share the interest of the reviewer – we would like to point out that those two research questions have never been part of those we wanted to address with out manuscript. Specifically, as far as the first point is concerned (*i.e.*: that adversarial purification is a viable general recipe to obtain adversarial robustness), we would like to point out that [2] already provides very significant evidence in such sense. As far as the latter point is concerned (*i.e.*: that *naïve* image denoising is significantly different from adversarial purification), [1] already provides such results.
>      In our work, we wanted to address a different question, *i.e.* whether it is possible to integrate adversarial training and adversarial purification in a mutually beneficial way, providing tangible benefits *w.r.t.* each of the separate methods alone. We would kindly ask our paper to be evaluated as such.
> 2.   As far as the scope of our paper is concerned, we would ask the reviewer – as well as any of its readers – to stick to the experimental setup we defined, and to the eventual future developments of our work we outlined – without unduly generalising our results.
>      As far as a hypothetical comparison of `CARSO+AT(baseline)`  and `CARSO+MART/AWP/SCORE/ADML` is concerned, we believe it not to be an adequate test to verify our claims. Indeed, in the eventuality – as the reviewer suggests – that `CARSO+MART/AWP/SCORE/ADML` is in fact **stronger** than `CARSO+AT(baseline)`, no specific conclusions can be reached about the usefulness of having added CARSO to the mix. At best, the result could prove that – regardless of it being used alone, or as part of a CARSO-like defence scheme – `MART/AWP/SCORE/ADML` are better training protocols than `baseline`. Even though we are potentially interested in an investigation of such kind, such research question is orthogonal to that addressed by our paper.

---

> ### Author Response · Authors · 2023-11-23
> **Comment to the post-rebuttal review of Reviewer VTuG (part 2 of 2)**
>
> 3.   With respect to suggestions of a comparison between `A+MART/AWP/SCORE/ADML` and `CARSO+MART/AWP/SCORE/ADML` (with `A` being a generic *state-of-the-art* adversarial purification model), similar considerations do apply. Additionally, a more technical reason should be also considered. In fact, CARSO is not a conventional adversarial purification model – *i.e.* one that directly maps a *perturbed* input to a *clean* one. We refer to the introduction part of this comment for a more thorough explanation of what CARSO *is* and *is not*.
>      As a consequence, this would not allow for direct use of existing (*SotA* or otherwise) adversarial purification models to take the place of the *purifier* within CARSO, in an eventual comparison of the kind suggested by the reviewer. On the other hand, the surreptitious use of the architecture of an existing *input-to-input* conventional *purification-based* defence to be re-trained as the purifier of CARSO would defy the very meaning of *SotA* – such architecture being employed for a task it was never meant to solve.
>      Our experiments, instead, show that the `B+CARSO` defence provides better robust accuracy as opposed to `B` alone, with `B` being a generic *state-of-the-art* AT-based defence. Moreover, the same `B+CARSO` defence is able to overcome also *state-of-the-art* adversarial purification models.
>
> As a conclusive comment, we noticed that the *Reviewer* updated the evaluation of our work from 3 (*reject*) to 1 (strong reject) after our rebuttal – without providing a direct justification for such choice. We are left quite perplexed and dubious, and would like to ask the *Reviewer* for a more precise and detailed explanation.
>
> ---
>
> ### References
>
> [1] Shixiang Gu and Luca Rigazio. Towards deep neural network architectures robust to adversarial examples. In Workshop Track of the International Conference on Learning Representations, 2015.
> [2] Huanran Chen, Yinpeng Dong, Zhengyi Wang, Xiao Yang, Chengqi Duan, Hang Su, and Jun Zhu. Robust classification via a single diffusion model, 2023.

---

> > ### Comment · Reviewer_VTuG · 2023-11-23
> > **More Explanation**
> >
> > The authors did not understand what really contributional point for the adversarial area is. We do not want to know the question: **is it possible to integrate adversarial purification and adversarial robustness?**. This point is not contributional and not novel thought. If it is novel, then combining all of things with adversarial will be novel, but in this world nobody want this type of research without any insight. For example, Object Detection + Adversarial, Segmentation + Adversarial, Lane Detection + Adversarial, Medical + Adversarial, Langugae + Adversarial,  LLM + Adversarial. Why is this important?
> >
> > From this reason, the reviewer strongly recommends the authors should make the critical problem settings, while re-organizing logic flow very differently because the model is so simple and not contributional and experiments are not enough, though the authors think it is enough and too much evidential. If my opinion is wrong, then this paper's score distribution is inclined to be positive above accept,  however, it is not. The reviewer thinks this paper cannot be accepted until my suggentions are applied and if it is re-submission due to rejection before, the reviewer thought this paper did not try to improve all of quality and critical issue.

---

> > > ### Author Response · Authors · 2023-11-23
> > > **Comment on “More Explanation” by Reviewer VTuG**
> > >
> > > We thank the Reviewer for the clarification on his/her opinion about our paper. However, we would like to notice how such answer does not address the reason behind the *post-rebuttal* downgrade of the paper review score. Additionally, the Reviewer keeps iterating in providing new, incrementally different reasons for rejection – making it difficult to organise thoughts along a clear, consistent line of argumentation.
> > >
> > > With respect to the merit of the matter, we find the Reviewer assertion *“The authors did not understand what really contributional point for the adversarial area is. We do not want to know the question: is it possible to integrate adversarial purification and adversarial robustness?”* to be strongly opinion-based, and devoid of clear scientific justification.
> > >
> > > We agree with the Reviewer that the plain juxtaposition of two – potentially unrelated – fields or techniques (*e.g.* `adversarial robustness + (something else)`) does not warrant, on its own, the degree of novelty and interest from the community required for publication. However, he opine that presenting our work as such is strongly deceiving.
> > >
> > > Not only our work does not randomly blend two generic fields as the reviewer seems to convey by his/her examples (*Object Detection + Adversarial, Segmentation + Adversarial, Lane Detection + Adversarial, Medical + Adversarial, Language + Adversarial, LLM + Adversarial*) – but it also does not operate a simple juxtaposition of techniques at all.
> > > As explained many times, within the paper and in our rebuttals, CARSO proposes the novel method of *representation-conditional purification* as a way to improve adversarial robustness of the resulting system – in a way **that goes beyond the naïve idea of `purification+AT`**.
> > >
> > > As far as the relevance of our work – and the interest by the community – are concerned, we stand by the results we showed. We believe that – despite philosophical disagreements with the Reviewer about what constitutes ad *interesting* result – any method that produces, like ours, an **almost twofold increase on the current overall best CIFAR-100 $\ell_{\infty}$ robust classification accuracy** could potentially be of interest for the community. An explanatory insight (that the reviewer, curiously, laments the lack of) on the functioning of such method, and the reasons behind it, is provided in *Subsection 4.2* of our paper.

---

### Official Review · Reviewer_yvVv · 2023-10-31

**Soundness:** 2 fair
**Presentation:** 2 fair
**Contribution:** 2 fair
**Rating:** 5
**Confidence:** 3

**Summary:**

This paper proposed a method that incorporates two types of adversarial learning methodologies, adversarial training and adversarial purification, to improve adversarial robustness. The authors conducted experiments with the datasets CIFAR-10/100 to evaluate the adversarial robustness against AutoAttack and clean accuracy.

**Strengths:**

+ The C+AT/rand-AA improves over the baseline AT/AA, verifying the proposed method improved the adversarial robustness.

**Weaknesses:**

- Only evaluate the method on CIFAR-10 and 100, which are from the same image distribution. Should also evaluate other different datasets for comprehensiveness.

- Adversarially pre-trained models are used as the classifier in the proposed method. Should also compare with those methods following the same setting for fairness.

- The authors showed training times for different scenarios. It should be compared with other methods. Also, the comparison on inference time should also be given to evaluate the efficiency.

**Questions:**

1. Are there any other previous papers that have considered incorporating adversarial training and adversarial purification in a unified framework? If yes, please list them.

2. The authors mentioned they draw inspiration from neurocognitive processes underlying cued recall and recognition. Please elaborate a little bit more details about this neurocognitive process and why it inspires the proposed method.

---

> ### Author Response · Authors · 2023-11-16
> **Rebuttal to Official Review by Reviewer yvVv (part 1 of 2)**
>
> We thank the Reviewer for his/her comments and suggestions.
>
> **On scaling to more complex datasets**
> We do agree with the Reviewer that extending the experimental assessment of our method to other datasets beyond CIFAR-10/100 (*e.g.* scaling up to ImageNet) would constitute a valuable extension to out work – and we are actively working towards such end.
> However, we would like to restate once more that our work is fundamentally concerned with whether it is possible to blend *adversarial training* and *adversarial purification* by conditioning the purification process on the internal representation of the classifier. With no prior work following such line of thought – before trying to *scale up* – we have been careful to ensure significant conclusions may be reached from a relatively simple, *proof-of-concept* set of experiments.
> Indeed, we deliberately choose a VAE as a purification model – whose use as an adversarial purifier has been long abandoned due to its relatively scarce expressiveness in image reconstruction – to allow for exact differentiability of the sampling process, and to emphasise that it is the **interplay with the AT-trained classifier** that is able to strongly improve robustness; and not the exploitation of an already *SotA* purification approach. In such constrained setting, and with no significant risk of robustness overestimation (as would instead happen when gradient approximations are used [6]) we still manage to overcome both state-of-the-art *AT-based* models and diffusion-based *purification* methods.
> A *scale-up* of our approach is indeed possible – both in terms of data size and task complexity. Reasonably doing so would require the use of a more modern and capable purification architecture (*e.g.* *score-*, *diffusion-* or *flow-* based), the complete rework of the experimental assessment (since adversarial attacks against such generative model are an active field of research – and doubts have been cast around the adequacy of AutoAttack in such sense), and significant engineering contributions. We defend that including such more complex scenario in our work, at this stage, would damage clarity of our contribution.
>
> **On comparison to adversarially pre-trained classifier models**
> The reviewer is correct in noticing that adversarially pre-trained models are used as part of the proposed method (CARSO). The comparison of CARSO with those models in terms of both clean and AutoAttack $\ell_{\infty}$ adversarial accuracy is already contained in *Table 2* (columns 4 and 5 for clean accuracy; columns 6 and 7 for robust accuracy).
> Following the evaluation guidelines of RobustBench [2] – *i.e.* considering AutoAttack as a standardised proxy for worst-case adversarial robustness against unforeseen attacks – models not endowed with stochastic components or non-deterministic forward passes (*i.e.* the original models) have been tested with the deterministic version of AutoAttack. The derived CARSO counterpart – being stochastic in its forward pass – has been tested against the stochasticity-aware version of AutoAttack, the rest of the setup being the same. Testing of CARSO against the standard version of AutoAttack always results in a much overestimated robustness, whereas testing the original models with its stochasticity-aware version is both unnecessary and still results in overestimated robustness.
>
> Overall, we do believe that the choices operated as part of our comparison allow to reflect best the assessment goals for which AutoAttack has been originally devised, and is used by the robust DL community.
>
> **On the comparison of training and inference times**
> We totally agree with the Reviewer that a fair comparison with training and inference times of other methods would improve the strength and presentation of the efficiency of our proposal (which is, however, a marginal aspect of the whole contribution).
>
> In any case, a fair comparison would require strong homogeneity in the hardware setup – almost always preventing reliance on the times shown in the respective papers, for such purpose. Given also the large computational budget required to re-train models for the sake of such comparison, and the relatively marginal role of such precise efficiency measurements in the context of safe and adversarially-robust DL, the large majority of similar published papers in the field only report elapsed training time, and a precise description of the underlying hardware. We uniformed our contribution to such common practice.

---

> ### Author Response · Authors · 2023-11-16
> **Rebuttal to Official Review by Reviewer yvVv (part 2 of 2)**
>
> **On prior art related to the blending of adversarial training and purification**
> As stated in the 5th paragraph of the *Introduction*, we are genuinely not aware of any previous work that integrates *adversarial training* and *adversarial purification* in an organic, mutually-beneficial way.
>
> Of course, one may think of the stacking of a *purifier* and an *adversarially-trained* classifier as a simple technique to leverage both kinds of adversarial defences. However, such setup would expose itself to known vulnerabilities [1] resulting in generally worst robustness compared to the adversarially-trained classifier alone.
>
> **On the neurocognitive inspiration of CARSO**
> In the *Introduction* of our paper, we state that our method has been inspired by the *neurocognitive processes underlying cued recall and recognition*. We would like to reiterate that such consideration was neither meant to be an analogical explanation, nor that our method is endowed by some kind of neurobiological plausibility: the works mentioned right after (*i.e.* [3], [4] and [5]) provided a suggestion towards the development of our main contribution: the *representation-conditional* generation of (purified, in our case) images.
> Similarly to how a biological stimulus (of same or different modality) can evoke a neuronal ensemble whose projection onto another ensemble may directly influence a collateral task – the whole process of image purification and subsequent classification (cued recognition) within CARSO is conditioned by the evoked internal representation within the classifier, when it is fed the original, potentially perturbed, input (cue). As part of such process, additionally, the actual classification does not occur directly on the original input (cue), but on a *version* of it that is *recalled* from the results of *experience* (the training of the VAE), idealised in its form (in our case: purified from perturbations) – *i.e.* constituting a *cued recall* of the original.
>
> ---
>
> ### References
>
> [1] Shixiang Gu and Luca Rigazio. Towards deep neural network architectures robust to adversarial examples. In Workshop Track of the International Conference on Learning Representations, 2015.
> [2] Francesco Croce, Maksym Andriushchenko, Vikash Sehwag, Edoardo Debenedetti, Nicolas Flammarion, Mung Chiang, Prateek Mittal, and Matthias Hein. RobustBench: a standardized adversarial robustness benchmark. In Thirty-fifth Conference on Neural Information Processing Systems Datasets and Benchmarks Track (Round 2), 2021.
> [3] Endel Tulving and Donald M. Thomson. Encoding specificity and retrieval processes in episodic memory. In Psychological Review, 1973.
> [4] James Eric Eich. The cue-dependent nature of state-dependent retrieval. In Memory & Cognition, 1980.
> [5] J.J. Medina. The biology of recognition memory. In Psychiatric Times, 2008.
> [6] Huanran Chen, Yinpeng Dong, Zhengyi Wang, Xiao Yang, Chengqi Duan, Hang Su, and Jun Zhu. Robust classification via a single diffusion model, 2023.

---

### Official Review · Reviewer_8cBw · 2023-11-01

**Soundness:** 3 good
**Presentation:** 2 fair
**Contribution:** 3 good
**Rating:** 5
**Confidence:** 4

**Summary:**

This work proposes adversarial defense to natural image classifiers by blending adversarial training with adversarial purification. Besides, a bag of tricks were used to improve defensive capability. Experiments were also conducted on CIFAR-10 and CIFAR-100 to show its effectiveness.

**Strengths:**

- The studied problem and the proposed method are interesting and the paper is easy to follow.
- Experiments in Table 2 showed that the robust accuracy of the proposed method is competitive with state-of-the-art methods.

**Weaknesses:**

- I'm not sure whether the defensive effectiveness comes from the combination of adversarial purification and adversarial training or the bags of tricks utilized in this work.
- In Table 2, please explicitly mention the state-of-the-art method and its reference rather than vaguely refer to sota in RobustBench as the sota can change with time, and it's unclear whether it's fair to compare ith sota as the experimental settings (architecture, training setups may vary).
- Besides robust accuracy, why not compare with sota regarding clean accuracy, while in the present form, it seems the authors only reported comparisons with standard adversarial training (2018).

**post rebuttal**

I have carefully checked the rebuttal and comments from the other reviewers, and I still believe this paper can be much further improved particularly in terms of the presentation (as the readers might be confused by some presented results, contributions, or tricks).

**Questions:**

See weakness.

**Details Of Ethics Concerns:**

NA.

---

> ### Author Response · Authors · 2023-11-16
> **Rebuttal to Official Review by Reviewer 8cBw (part 1 of 2)**
>
> We thank the Reviewer for his/her comments and suggestions.
>
> **On the *bag of tricks* used in this work**
> We would like to start by pointing out that the *bag of tricks* used in our proposal was **not** motivated by the goal of improving the defensive capability of the resulting model, as the reviewer seems to assume – and was never described as such in any passage of the paper.  Such tricks (see *e.g.* the bullet list part of the *Introduction*) have been devised to **allow** the actual training of the model (which would have been impossible otherwise, *e.g.* for computational reasons), to **accelerate** it, and to **stabilise** it – sometimes even sacrificing robust accuracy.
>
> More in detail:
>
> -   The *example and internal representation pre-encoding* trick was motivated by the need to tailor the compressive process (of *examples* and *internal representations* respectively) to the specific structure of respective data types, while also avoiding the use of a very large and deep, indiscriminately fully-connected encoder. Since such requirements had to be matched to the principles of operation of a VAE, a stacked encoders structure was finally adopted. *Section 4.3* describes such reasoning in much richer detail, and never mentions defensive capabilities.
>
> -   The *instance specific tuning* trick(s) can be further subdivided into two:
>
>     -   The choice of decoder architecture was again justified by the type of data (natural images) to be generated, and to the need not to introduce gratuitous lossy artifacts in the reconstructions. This practice can be assumed to be quintessential to any reasonable use of deep learning, and not specific to adversarial defences.
>     -   The use of *internal representation* subsets, instead, actually **decreases** defensive capabilities of the method (as we observed, in the incipient phases of our work, from experiments with smaller networks), but was necessary to ensure scalability regardless of the size of the (representation of the) *classifier*.
>
>     Details on both such sets of tricks are contained in *Section 4.5*, with no mention of an increase in defensive effectiveness.
>
> - As far as the *adversarially balanced batches* trick is concerned – even though its net result is an increase in the resulting robust accuracy (relatively limited, though necessary to reliably overcome the state of the art) – it was developed as a way to stabilise the later phases of training. Specifically, given the need to train the purifier on both *corrupt-to-clean* and *clean-to-clean* mapping, some different example ordering schemes were explored: sequentially in homogeneous batches, determining uniformly at random whether each point had to be perturbed or not, and detailedly balancing each batch. Since the final (both clean and robust) accuracy of the resulting model depended significantly on the composition of the last training batches – in the first two cases – either an approach based on validation-driven early stopping, or based on detailed batch composition was needed. Validation of adversarial robustness would have in turn depended on the specific composition of the validation set; as a consequence, the *trick* in its final form was employed – removing the problem at its roots.
>
> **On the origin of defensive effectiveness**
> Given the explanation contained in the previous paragraph, we defend that it is indeed the blending of *adversarial training* and *representation-conditional adversarial purification* the reason for the observed increase in robust accuracy. The overall effect of the *bag of tricks* could instead be estimated from slightly positive at best (due to *adversarially balanced batches*) to significantly negative at worst (due to *internal representation subsetting*).
>
> **On the *SotA* results referred to**
> We apologize for the lack of clarity *w.r.t.* the exact sources of *SotA* results referred to both in *Table 2* and *Section 5.2*. It was the furthest to our intention to *conceal* such sources by referring to positions of the *RobustBench* leaderbord – which, alone, we understand may be source of confusion.
>
> While the CIFAR-10 *purification-based* *SotA* source is mentioned right underneath *Table 2* (*i.e.* [1]), the remaining AT-based *SotA* results for CIFAR-10 ([2]) and CIFAR-100 ([3]) were only mentioned as such across the *Introduction* and *Related Work* sections.
>
> We will surely update the section and the table to make more explicit and clear references.
>
> For the sake of completeness, and future reference, they are once more reported below in bullet-list form:
>
> -   CIFAR-10, AT-based *SotA*, refers to [2];
> -   CIFAR-10, purification-based *SotA*, refers to [1];
> -   CIFAR-100, AT-based *SotA*, refers to [3].

---

> ### Author Response · Authors · 2023-11-16
> **Rebuttal to Official Review by Reviewer 8cBw (part 2 of 2)**
>
> **On the *SotA* for clean accuracy**
> To better pinpoint the observation, we would kindly ask the Reviewer to clarify whether the *SotA* clean accuracy referred to was intended to mean the *clean* accuracy of the best (*SotA*) **robust** models – or the general *SotA* clean *accuracy* on the respective dataset, regardless of robustness constraints.
>
> Assuming that the reviewer refers to the former interpretation – we could easily add a further column in *Table 2* conveying such information. In any case, we would point out that, at the moment of the writing, clean accuracy of robust *SotA* models (see previous answer for precise references) are within 0.5% accuracy *w.r.t.* the best baselines in *Table 2*.
>
> ---
>
> ### References
>
> [1] Huanran Chen, Yinpeng Dong, Zhengyi Wang, Xiao Yang, Chengqi Duan, Hang Su, and Jun Zhu. Robust classification via a single diffusion model, 2023.
> [2] ShengYun Peng, Weilin Xu, Cory Cornelius, Matthew Hull, Kevin Li, Rahul Duggal, Mansi Phute, Jason Martin, and Duen Horng Chau. Robust principles: Architectural design principles for adversarially robust CNNs, 2023.
> [3] Zekai Wang, Tianyu Pang, Chao Du, Min Lin, Weiwei Liu, and Shuicheng Yan. Better diffusion models further improve adversarial training, 2023.
> [4] Weili Nie, Brandon Guo, Yujia Huang, Chaowei Xiao, Arash Vahdat, and Anima Anandkumar. Diffusion models for adversarial purification. In Proceedings of the International Conference on Machine Learning, 2022.

---

> ### Author Response · Authors · 2023-11-23
> **Comment to the post-rebuttal review of Reviewer 8cBw**
>
> We thank the *Reviewer* for his/her *post-rebuttal* comment. We also apologise for the initial confusion caused by the lack of notification about the publishing of such comment – having it been written as an edit to an already-published review instead of a new comment, necessary to ensure a notification to the *Authors*.
>
> As far as specific points of merit are concerned, we have slightly updated our manuscript – to integrate some clarifying details as suggested by the original reviews. With respect to all other issues identified, we believe our work, especially in its *post-rebuttal* form – not to be misleading to the readers – in any of the specific aspects mentioned (*i.e.* results, contributions, tricks). We refer to the specific rebuttals for an in-depth discussion on the merits of the matter.

---

### Official Review · Reviewer_T2NZ · 2023-11-03

**Soundness:** 2 fair
**Presentation:** 2 fair
**Contribution:** 2 fair
**Rating:** 5
**Confidence:** 2

**Summary:**

- Draft presents a method for defending against adversarial attacks
- Specifically, the proposed method maps the feature representations of (normal or adversarial) inputs (extracted by an adversarially trained classifier) to a sample of tentatively clean reconstructions. This mapping is realized via a conditional variational encoder (VAE).
- An estimate of the clean inference is obtained by fusing the inference of adversarially trained classifier on multiple reconstructed features (o/p of VAE)
- Experiments are performed on CIFAR-10 and CIFAR-100 datasets.
- Better results are reported than other methods compared against.

**Strengths:**

- The idea of generative purification from adversarially trained feature representations - despite being a new combination of existing ideas - appears interesting in the context of adversarial defense.

**Weaknesses:**

- Generative model-driven purification as a defense against adversarial attacks has been established to a reasonable extent. Existing works demonstrated the effectiveness of this framework using denoising autoencoders (DAE), UNet-based DAE, GAN, VAE, etc. (section 2 of the draft provides references). However, scaling to complex datasets such as ImageNet has been challenging in this context. This draft restricts the experimental analysis only to simpler datasets (CIFAR). Since sophisticated models that are adversarially trained on ImageNet dataset are available easily, readers would expect the draft to experiment with them too.
- Clarity of the framework description needs slight improvement (please refer to the questions section of the review).
- It is not discussed clearly in the draft what the improvement in the proposed method is compared to existing purification methods. The draft claims that their method adds a smoothness penalty to the reconstruction loss. However, not much has been discussed on it.

**post rebuttal**
- The reviewer thanks the authors for a detailed response. The reviewer has carefully read all the other reviews and corresponding responses.
- The reviewer strongly opines that the draft needs to be improved w.r.t. the clarity of presentation clarifying all the issues raised by the reviewers. Moreover, the discussion and the experiments presented do not strongly convince about the effectiveness of the proposed framework against adversarial attacks. The reviewer reckons that the authors' notion of a proof-of-concept level of work needs to be substantially supported (with the points mentioned by the reviewers) to be discussed at a venue such as ICLR.

**Questions:**

- The draft reads that to sample from the generative model in the proposed framework, the auxiliary encoder $\mathcal{D}$ is unnecessary. However, it appears that the sampling process needs the encoder's output - which is driven by both the auxiliary encoders $\mathcal{C}$ and $\mathcal{D}$ -  concatenation of $z_i$ and $c'$. Authors may provide a clarification for this.
- In section 4.5, it is mentioned that the proposed approach - instead of including the conditioning on tensor, which is customary to the conditional VAEs- designed a DGDN-based decoder. However, the schematic representation in Figure 1 depicts the decoder being conditioned on the output of the auxiliary encoder $\mathcal{C}$. Authors may provide clarity on this discrepancy.
- How is it different to learn the purifier from the feature space of an adversarial trained classifier than learning from that of a normally trained classifier?

---

> ### Author Response · Authors · 2023-11-16
> **Rebuttal to Official Review by Reviewer T2NZ (part 1 of 2)**
>
> We thank the Reviewer for his/her comments and suggestions.
>
> **On the novelty of our work**
> Firstly, we would like to clarify the scope and elements of novelty of our contribution. As the Reviewer asserts, and as we transparently acknowledge in the paper, both the ideas of *adversarial training* and *adversarial purification* are not novel to our proposal. However, our method goes beyond the simple merging of the two paradigms (as would be, instead, the case of a simple *stacking* of a purification model and an adversarially-trained classifier). Indeed, there exist known settings in which such simple juxtaposition produces an overall decrease in *end-to-end* robustness compared to its parts alone [1]. Instead, our paper is the first to propose the idea of *representation-conditional* purification, unlocking a new mechanism to leverage robust classification by the principles explained in *Section 4.2*.
>
> **On scaling to more complex datasets**
> We do agree with the Reviewer that extending the experimental assessment of our method to other datasets beyond CIFAR-10/100 (*e.g.* scaling up to ImageNet) would constitute a valuable extension to out work – and we are actively working towards such end.
> However, we would like to restate once more that our work is fundamentally concerned with whether it is possible to blend *adversarial training* and *adversarial purification* by conditioning the purification process on the internal representation of the classifier. With no prior work following such line of thought – before trying to *scale up* – we have been careful to ensure significant conclusions may be reached from a relatively simple, *proof-of-concept* set of experiments.
> Indeed, we deliberately choose a VAE as a purification model – whose use as an adversarial purifier has been long abandoned due to its relatively scarce expressiveness in image reconstruction – to allow for exact differentiability of the sampling process, and to emphasise that it is the **interplay with the AT-trained classifier** that is able to strongly improve robustness; and not the exploitation of an already *SotA* purification approach. In such constrained setting, and with no significant risk of robustness overestimation (as would instead happen when gradient approximations are used [2]) we still manage to overcome both state-of-the-art *AT-based* models and diffusion-based *purification* methods.
> A *scale-up* of our approach is indeed possible – both in terms of data size and task complexity. Reasonably doing so would require the use of a more modern and capable purification architecture (*e.g.* *score-*, *diffusion-* or *flow-* based), the complete rework of the experimental assessment (since adversarial attacks against such generative model are an active field of research – and doubts have been cast around the adequacy of AutoAttack in such sense), and significant engineering contributions. We defend that including such more complex scenario in our work, at this stage, would damage clarity of our contribution.
>
> **On the comparison with existing *adversarial purification* models**
> As described earlier on, the core methodological novelty of our method *w.r.t.* existing purification-based defences lies in the idea of the purification process being conditional on the internal representation of an adversarially-trained classifier. As said, this allows to exploit the behaviour of the gradient at the level of classifier neurons suggested in *Section 4.2*.
> From a purely empirical viewpoint, our contribution is able to significantly improve upon existing purification-based methods – exceeding the current *state of the art* in terms of $\ell_{\infty}$ robust accuracy against `AutoAttack`. In particular (see *Table 2*), our best model obtains $0.7570$ robust accuracy on CIFAR-10. This compares to the current best obtained by purification on the same dataset: $0.7324$ [2].
> Finally, gradient-based robustness evaluation of existing purification methods relies on gradient estimates through the model (*e.g.* via the adjoint method, as in [3]) due to the computational infeasibility of its exact computation. This may lead to inaccurate estimates and, in turn, an overconfident evaluation. Our models (with the deliberate choice of a VAE as purifier, which hampers overall purification ability due to scarce model expressiveness, but allows fast *end to end* exact differentiability) uses the exact gradient. Of course, one may use different, more capable purification models within the framework of CARSO: this would allow even increased adversarial robustness, but clearly complicates a fair assessment of robustness, and introduces the need of a radical rework of the experimental setup.

---

> ### Author Response · Authors · 2023-11-16
> **Rebuttal to Official Review by Reviewer T2NZ (part 2 of 2)**
>
> **On the *smoothness penalty***
> In no passage of the paper it is claimed that **our proposed method (CARSO)** adopts a smoothness penalty to the reconstruction loss. We always use *summed pixel-wise channel-wise* binary cross entropy (on $[0,1]$-shrunken inputs) as the reconstruction component of our VAE loss (see *Section 5.1*, *Outer minimisation*, paragraph 2 – and *Table 9*). The only passage where a *smoothness penalty* is mentioned is right after a direct reference to related work [1] where a smoothness penalty is proposed, and to which the passage refers to. Additionally, given that the section is dedicated to (and titled) *Related Work*, we do not believe such passage may mislead readers to believe such proposal is ours. We will make the phrasing furtherly explicit in a new version of our draft.
>
> **On whether the auxiliary *data pre-encoder* is required at inference time**
> We refrain that inference-time operation of CARSO **does not** require the auxiliary *data pre-encoder* $\mathcal{D}$, as stated in the manuscript, and correctly understood by the Reviewer. Indeed, not even the *proper* encoder is needed: only the auxiliary *representation pre-encoder* $\mathcal{C}$ is necessary.
> As a matter of fact, at inference time, $c^{\prime}$ is obtained by feeding the extracted internal representation from the classifier to $\mathcal{C}$, whereas the vectors $z_i$ are sampled from the *i.i.d.* Standard Normal priors of the VAE. This is explained in *Section 4.1*, *Inference* and in *Figure 1* right next to it.
> Such design choice is crucial to ensure that no gradient flows from the input to the bottleneck of the VAE during an adversarial attack, preventing known vulnerability pathways to be exploited [1].
>
> **On the role of $c^{\prime}$ in the reconstruction task**
> In *Section 4.5*, we **do not** state that the reconstruction task is not conditional on the conditioning tensor $c^{\prime}$. As it appears from *Figure 1*, and as the Reviewer notices, such reconstruction is indeed conditional on $c^{\prime}$. What we remark instead is the absence of such tensor $c^{\prime}$ **from the target** of the reconstruction task.
>
> **On the use of non-adversarially-trained classifiers**
> Within the incipient phases of our work, we experimented with similar architectures where the classifier was trained on clean inputs only. In none of such situations (except on simpler, smaller datasets) we obtained better or comparable end-to-end adversarial accuracy, with results always significantly better *w.r.t.* the undefended model – but largely varying on a case by case basis. When a *cleanly-trained* classifier was used also for the classification of reconstructed inputs, decreased *clean* accuracy was sometimes observed – due to increased sensitivity of the classifier to noise or reconstruction artifacts. Since an adversarially-trained classifier was deemed necessary, in the end, for the classification of reconstructed data, subsequent efforts focused in the direction of the final architecture proposed. In such setting, the classifier is used also as a *featuriser* – allowing to exploit during inference the peculiarities of gradients arising as part of an adversarial attack described in *Section 4.2*. Additionally, such choice increases *end-to-end* robustness, and a more consistent *clean* accuracy across trials.
>
> ---
>
> ### References
>
> [1] Shixiang Gu and Luca Rigazio. Towards deep neural network architectures robust to adversarial examples. In Workshop Track of the International Conference on Learning Representations, 2015.
> [2] Huanran Chen, Yinpeng Dong, Zhengyi Wang, Xiao Yang, Chengqi Duan, Hang Su, and Jun Zhu. Robust classification via a single diffusion model, 2023.
> [3] Weili Nie, Brandon Guo, Yujia Huang, Chaowei Xiao, Arash Vahdat, and Anima Anandkumar. Diffusion models for adversarial purification. In Proceedings of the International Conference on Machine Learning, 2022.

---

> ### Author Response · Authors · 2023-11-23
> **Comment to the post-rebuttal review of Reviewer T2NZ**
>
> We thank the *Reviewer* for his/her *post-rebuttal* comment. We also apologise for the initial confusion caused by the lack of notification about the publishing of such comment – having it been written as an edit to an already-published review instead of a new comment, necessary to ensure a notification to the *Authors*.
>
> As far as points of merit are concerned, our definition of the experiments and results contained in our work as *proof-of-concept* pertained to a specific comment of *Reviewers* with respect to the choice of benchmark datasets utilised in the paper (*i.e.* CIFAR-10 and CIFAR-100) – and definitely not to the strength and value of such results.
> Indeed, within the experimental scope so identified, our results show a clearly improved $\ell_{\infty}$ adversarial accuracy of CARSO against any of the baselines considered, and also against the current best published *adversarial training* and *adversarial purification* models and protocols.
>
> Even more so, because of the fact that – as mentioned in both global and specific rebuttals – our choice of experimental framework and purification model to be employed, was not driven by purification performance. Instead, exact differentiability of the purification model, and constraints on computational expenditure, were mainly took into consideration – even at the cost of decreased overall robustness, and a much harder setup for our model to defend.
>
> With respect to all other issues identified, we believe our work, especially in its *post-rebuttal* form – not to be unclear or misleading to the readers – and refer to the specific rebuttals for an in-depth discussion on the merits of the matter.

---

### Author Response · Authors · 2023-11-16
**Rebuttal to Everybody**

We thank all Reviewers for their observations and suggestions on how to improve our manuscript. Specific points raised in individual reviews have been respectively addressed, *in depth*, as part of individual rebuttals. In the following, we will try to restate some core aspects of our contribution in the hope to foster a fairer, more contextualised evaluation of our work.

To begin with, we would like to clarify once more the main research question addressed by our work, and the core elements of novelty and contribution *w.r.t.* existing published literature.
Our research endeavour targets the following question: *Is it possible to integrate adversarial training and adversarial purification in a mutually beneficial way?* And the novel approach addressing such problem was found in *representation-conditional adversarial purification*, *i.e.* the interplay between an existing adversarially-trained classifier and a purifier acting on potentially-perturbed inputs, whose purification process is conditional on the internal representation of the classifier.

With no significant prior work to build upon, we focused on essential, *proof-of-concept*, solid experimentation. Our goal was ensuring that the effect of the integration of the two approaches was the clear reason for observed defensive capabilities – an not that of evaluation overconfidence or excessive reliance on existing *state-of-the-art* purification schemes. This choice justifies the use of a VAE as a purifier – that is exactly differentiable algorithmically, even though its performance on image purification is notoriously scarce. Similar in spirit is the use, as part of our experimental assessment, of $\ell_{\infty}$ AutoAttack – standardised and definitely not lacking comparative references – and the comparison with only the best-performing models in terms of robustness, at the moment of writing.

Within such experimental framing, the fact that we have been able to overcome the *state of the art*, both in comparison to *purification-* and *AT-*based defences – even though restricting ourselves to CIFAR-10/100 – proves our novel method to be built on solid grounds, with the *bag of tricks* proposed playing only a very marginal role in the robustness achieved – but nonetheless a crucial enabling factor to make training reliably converge **with limited computational budget**.

These results motivate efforts in the direction of scaling CARSO to more challenging benchmarks and use-cases. In such respect, adaptation of our method to ImageNet-like classification tasks would require a general reframing of our experimental setup. To be able to handle increased input size and a wider variability across more classes, the adoption of a more capable generative model (*e.g.* *diffusion-*, *score-* or *flow-*based) as purifier would be required. Additionally, with adversarial attack generation against such models being an open field of research, and AutoAttack possibly showing some of its limits, careful consideration of obtained results and their comparison to *SotA* becomes necessary.

A new version of our draft – occasionally mentioned in individual rebuttals, integrating some minor clarificatory suggestions – is underway. We will upload it after the initial phase of active discussion, to allow for the integration of eventual further edits and corrections – relieving Reviewers from the need to manage many incremental `pdfdiff`s.

---

### Comment · Area_Chair_rd6z · 2023-11-20
**Comments on Authors' responses**

Dear Reviewers, The authors have responded to your valuable comments. Please take a look at their responses!

---

### Author Response · Authors · 2023-11-22
**Comment on the Rebuttal Revision**

In the absence of further feedback from the Reviewers, we have just uploaded the *rebuttal revision* of our manuscript. The content of such modifications is contained in our rebuttal to the original reviews.

The summary of modifications is the following:

-   Added further clarification that the *addition of a smoothness penalty* to VAE-based adversarial purification was proposed by *Gu and Rigazio (2015)*;
-   Added explicit references to the *state-of-the-art* models referred to in *Table 2* and *Subsection 5.2*;
-   Added a paragraph within the *Conclusion* further detailing requirements and challenges of scale-up and testing of our method on larger and more complex datasets;
-   Fixes to capitalisation and specification of book chapters for some *References*.

**Addendum**
We apologise for the initial confusion caused by the lack of notification about the publishing of *post-rebuttal* comments by the Reviewers – having them been written as an edit to an already-published review instead of a new comment, necessary to ensure a notification to the *Authors*.

We have further addressed the specific points raised in *post-rebuttal* comments.

---

### Meta-Review · Area_Chair_rd6z · 2023-12-05

**Metareview:**

In this paper, the authors studied how to blend both adversarial training and purification to improve adversarial robustness.
During the rebuttal period, the authors have tried to address the comments from the reviewers in order to clarify the proposed method.
However, at least two reviewers have read the rebuttal from the authors, and they still commented that the paper needs to be clearly clarified, and the explanation and experimental results are still not convincing.
This work, at its current status, is not accepted.

**Justification For Why Not Higher Score:**

The authors' rebuttal did not satisfy the reviewers. The results are not convincing and not fully supported by sufficient evidences.

**Justification For Why Not Lower Score:**

the authors have addressed some comments from the reviewers, though they do not reach a consensus.

---

### Decision · Program_Chairs · 2024-01-16

Reject